# Structure of tick-borne encephalitis virus and its neutralization by a monoclonal antibody

Tibor Füzik [1], Petra Formanová[2], Daniel Růžek [2,3], Kentaro Yoshii[4], Matthias Niedrig[5] & Pavel Plevka[1]

Tick-borne encephalitis virus (TBEV) causes 13,000 cases of human meningitis and encephalitis annually. However, the structure of the TBEV virion and its interactions with antibodies are unknown. Here, we present cryo-EM structures of the native TBEV virion and its complex with Fab fragments of neutralizing antibody 19/1786. Flavivirus genome delivery depends on membrane fusion that is triggered at low pH. The virion structure indicates that the repulsive interactions of histidine side chains, which become protonated at low pH, may contribute to the disruption of heterotetramers of the TBEV envelope and membrane proteins and induce detachment of the envelope protein ectodomains from the virus membrane. The Fab fragments bind to 120 out of the 180 envelope glycoproteins of the TBEV virion. Unlike most of the previously studied flavivirus-neutralizing antibodies, the Fab fragments do not lock the E-proteins in the native-like arrangement, but interfere with the process of virus-induced membrane fusion.

[1] Structural Virology, Central European Institute of Technology, Masaryk University, Kamenice 753/5, 62500 Brno, Czech Republic. [2] Department of Virology, Veterinary Research Institute, Hudcova 70, 62100 Brno, Czech Republic. [3] Institute of Parasitology, Biology Centre of the Czech Academy of Sciences, Branisovska 31, 37005 Ceske Budejovice, Czech Republic. [4] Laboratory of Public Health, Graduate School of Veterinary Medicine, Hokkaido University, Sapporo, 060-0818, Japan. [5] Centre for Biological Threats and Special Pathogens, Robert Koch Institute, Nordufer 20, 13353 Berlin, Germany. Correspondence and requests for materials should be addressed to P.P. (email: pavel.plevka@ceitec.muni.cz)

Tick-borne encephalitis virus (TBEV) infects a range of hosts including ruminants, birds, rodents, and carnivores that provide a reservoir from which the virus can be transmitted to humans[1]. Annually, Europe and Russia report 10,000–13,000 cases of TBEV-induced meningitis, encephalitis, or meningoencephalitis[2,3]. Mortality varies depending on the TBEV subtype. Whereas in Europe it is usually between 1 and 2%, with deaths occurring 5–7 days after the onset of the neurological symptoms[1], far-eastern TBEV causes more severe diseases with mortality in the range of 5–20%[2]. Long-lasting or permanent neuropsychiatric disorders are observed in 10–20% of infected patients[3].

Vaccines are an effective means of protection against flavivirus-caused diseases, including TBEV[4]; however, not all people in the TBEV-affected areas are vaccinated, and the numbers of annual infections are increasing[2]. Therefore, therapeutic tools against TBEV are required. One possible treatment is the infusion of neutralizing antibodies, which has been shown to confer protection against infection by several flaviviruses, including TBEV[5].

TBEV belongs to the family *Flaviviridae* of positive-sense, single-stranded RNA viruses. Mature flavivirus virions are membrane-enveloped with a diameter of 50 nm[6–8]. Attachment of the virus particles to cells is receptor-mediated, and the infection is initiated after the uptake of virions into endosomes[9,10]. Low pH in the endosomes initiates conformational changes of virus envelope glycoproteins that induce fusion of the virion membrane with the membrane of the endosome[11,12]. After delivery into the cytoplasm, the ~10,000-nucleotide-long positive-sense single-stranded RNA genome is translated into polyproteins that are co-translationally and post-translationally cleaved into functional subunits, which include protease, RNA-dependent RNA polymerase, capsid, and envelope proteins. Virus envelope (E) and pre-membrane (prM) proteins are co-translationally inserted into the membrane of the endoplasmatic reticulum. The replication of flaviviruses occurs in the cytoplasm, in close association with membranes, in the so-called virus replication factories[13]. Immature flavivirus virions are formed by budding of the complex of the genome with capsid proteins into the lumen of the endoplasmatic reticulum. The surface of the immature particles is covered with trimers of prM–E protein heterodimers[14–16]. The newly formed virions encounter acidic pH as they are transported into the Golgi complex and trans-Golgi network. The low pH induces reorganization of E-proteins into a herringbone-like arrangement[14], which starts from a nucleation center and then spreads around the particle[17,18]. Immature virions contain intact prM peptides that cover the fusion loops of the E-proteins and thus prevent fusion of the virus with intracellular membranes[19]. After the low-pH-induced reorganization of the envelope glycoproteins, a cleavage site for the protease furin within the prM peptide becomes exposed at the virion surface, and the peptide is cut into pr and M-fragments[14,20]. When the virions are released from cells into the extracellular space with neutral pH, the pr-peptides dissociate from the particles, rendering the virions mature and fusion-competent[14,21]. The structures of mature virions of the dengue (DENV), Zika (ZIKV), West Nile (WNV), and Japanese encephalitis viruses (JEV) and of the sub-viral particle of TBEV were solved previously by cryo-EM[6,7,22–24]. The structures of E-protein ectodomains of TBEV and other flaviviruses were determined in the form of dimers and post-fusion trimers[25–28].

Here, we report the structures of the native TBEV virion and its complex with the Fab fragments of the neutralizing antibody 19/1786. Our results indicate that the low-pH-induced protonation of histidines may contribute to disruption of the E–M heterotetramers and induce detachment of the E-protein ectodomains from the virus membrane. Furthermore, the binding of 19/1786

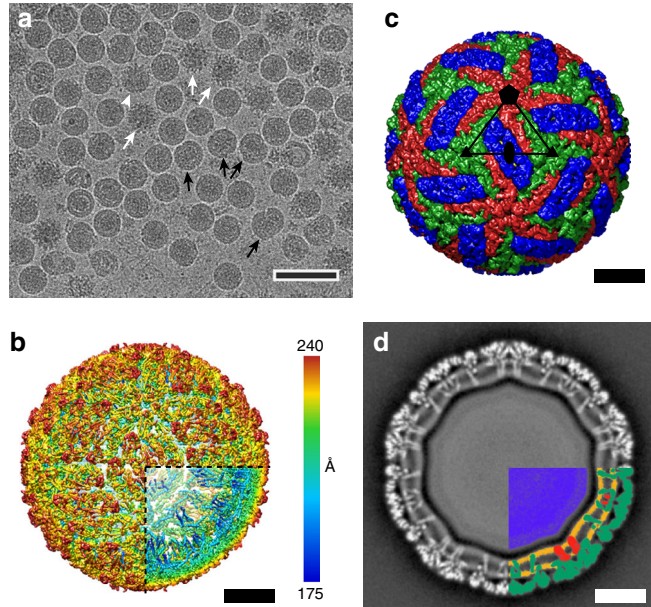

**Fig. 1** Structure of TBEV virion. **a** Cryo-EM image of TBEV virions. The sample contained mature, immature (white arrows), half-mature (white arrowheads), and damaged (black arrows) particles. Scale bar 100 nm. **b** B-factor sharpened electron-density map of TBEV virion, rainbow-colored according to distance from particle center. The front lower-right eighth of the particle was removed to show the transmembrane helices of E-proteins and M-proteins. **c** Molecular surface of TBEV virion low-pass filtered to 7 Å. The three E-protein subunits within each icosahedral asymmetric unit are shown in red, green, and blue. The three E-proteins in the icosahedral asymmetric unit form unique interactions with each other (for more details, see Supplementary Fig. 2). The black triangle shows the borders of a selected icosahedral asymmetric unit. **d** Central slice of TBEV electron density map perpendicular to the virus 5-fold axis. The virus membrane is deformed by the transmembrane helices of E-proteins and M-proteins. The lower right quadrant of the slice is color-coded as follows: nucleocapsid—blue; inner and outer membrane leaflets—orange; M-proteins—red; E-proteins—green. Scale bars in **b**, **c**, and **d** represent 10 nm

antibodies to the TBEV surface does not prevent the low-pH-induced movements of E-proteins; however, it does interfere with the virus-induced membrane fusion.

## Results and discussion

**Structure of TBEV virion.** Cryo-electron micrographs of purified TBEV virions showed smooth spherical particles with a diameter of 50 nm, similar to those of other flaviviruses (Fig. 1a)[6,8,23]. Many TBEV particles were irregular or damaged and therefore could not be used for cryo-EM reconstruction with icosahedral symmetry (Fig. 1a). Nevertheless, the structure of the mature TBEV particle was determined to a resolution of 3.9 Å (Fig. 1b, Supplementary Fig. 1a, b, c, Table 1). The quality of the map was sufficient to enable the building of the protein components of the TBEV envelope, which contains three E-proteins and three M-proteins in each icosahedral asymmetric unit. The surface of the TBEV virion is covered with small protrusions formed by glycans attached to the E-protein subunits (Fig. 2a). Two E-proteins and two M-proteins form a compact heterotetramer (Fig. 2b). Three of these heterotetramers constitute the so-called herringbone pattern characteristic of the envelopes of flaviviruses (Supplementary Fig. 1d)[6,23,29,30]. In contrast to the principles suggested by Caspar and Klug and unlike most isometric viruses, the three E-protein subunits within one icosahedral asymmetric unit form

**Table 1 Cryo-EM data collection, refinement, and validation statistics**

| | TBEV virion (EMD-3752) (PDB 5O6A) | TBEV-Fab 19/1786 (EMD-3754) (PDB 5O6V) | TBEV-Fab 19/1786, pH 5.8 (EMD-3755) |
|---|---|---|---|
| Data collection and processing | | | |
| Magnification | 75,000× | 75,000× | 75,000× |
| Voltage (kV) | 300 | 300 | 300 |
| Electron exposure (e⁻/Å²) | 22 | 22 | 22 |
| Defocus range (µm) | 0.8–3.8 | 0.8–3.7 | 0.8–3.4 |
| Pixel size (Å) | 1.063 | 1.063 | 1.063 |
| Symmetry imposed | Icosahedral | Icosahedral | Icosahedral |
| Initial particle images (no.) | 19,111 | 12,098 | 4515 |
| Final particle images (no.) | 11,882 | 5929 | 3831 |
| Map resolution (Å) | 3.9 | 3.9 | 19.2 |
| FSC threshold | 0.143 | 0.143 | 0.143 |
| Map resolution range (Å) | 3.8–6.2 | 3.8–7 | 19.2 |
| Refinement | | | |
| Initial model used (PDB code) | 1SVB, 3J27 | 1SVB, 3J27 | – |
| Model resolution (Å) | 3.9 | 3.9 | – |
| FSC threshold | 0.143 | 0.143 | – |
| Model resolution range (Å) | ∞–3.9 | ∞–3.9 | – |
| Map sharpening $B$-factor (Å²) | −115 | −104 | 0 |
| Model composition | | | – |
| Non-hydrogen atoms | 12,942 | 19,346 | – |
| Protein residues | 1689 | 2529 | – |
| Ligands | 3 | 3 | – |
| $B$-factors (Å²) | | | – |
| Protein | 116.2 | 154.1 | – |
| Ligand | 126.3 | 88.4 | – |
| R.m.s. deviations | | | – |
| Bond lengths (Å) | 0.006 | 0.007 | – |
| Bond angles (°) | 1.06 | 1.15 | – |
| Validation | | | – |
| MolProbity score (percentile) | 1.59 (100) | 1.66 (100) | – |
| Clashscore (percentile) | 2.62 (100) | 3.27 (100) | – |
| Poor rotamers (%) | 0.0 | 0.47 | – |
| Ramachandran plot | | | – |
| Favored (%) | 90.34 | 90.16 | – |
| Allowed (%) | 9.36 | 9.44 | – |
| Disallowed (%) | 0.30 | 0.40 | – |

unique interactions with the surrounding glycoproteins (Fig. 1c, Supplementary Fig. 2)[6–8]. Both E-proteins and M-proteins are anchored in the virion membrane, each by two trans-membrane helices (Fig. 2c, d). The inner and outer leaflets of the membrane are clearly separated in the cryo-EM map (Fig. 1d). However, individual lipids are not resolved in the reconstruction because of the fluidic character of the membrane. The shape of the virus membrane is not spherical; instead it closely follows the inner surface of the protein envelope. The membrane is deformed by insertions of the trans-membrane helices of E-proteins and M-proteins (Fig. 1d). Similar shapes of virion membranes were previously observed in DENV, ZIKV, and WNV[7,8,23]. Inside the envelope is a nucleocapsid core that is not ordered with icosahedral symmetry; therefore, the corresponding regions of the electron density map do not contain any resolved features (Fig. 1d, Supplementary Fig. 1c).

**Organization and structure of TBEV E-proteins**. The structures of TBEV E-proteins could be built for residues 1–492 out of 496. The E-protein is, according to the flavivirus convention, divided into four domains[27]. The three N-terminal domains are mostly composed of β-strands and form an ectodomain that covers the virion surface (Fig. 2c). Domain I, which has a β-barrel fold, constitutes the center of the ectodomain between the domains II and III (Fig. 2c). Domain I includes the only glycosylation site of TBEV at Asn154 (Fig. 2a). The cryo-EM reconstruction contains

densities corresponding to N-acetyl-D-glucosamine in all three E-protein subunits of the asymmetric unit. The E-proteins of the majority of TBEV, WNV, ZIKV, and JEV strains contain a single homologous glycosylation site, whereas that of DENV has an additional glycosylation site at Asn67[8]. It was shown that the glycosylation of the TBEV E-protein is important for the secretion of the virus from infected cells[31].

The domain IIs of two E-proteins, which form a dimer in the native TBEV virion, are in contact through an interface with a buried surface area of 1490 Å². The domain II contains a fusion loop formed by residues 100–109 with hydrophobic side chains. The loop is positioned at the tip of the ectodomain (Fig. 2a). It is essential for fusion of the virus membrane with that of an endosome, which enables delivery of the virus genome into the cytoplasm of a host cell. In mature TBEV, the loop is covered in a pocket formed by domains I and III of the other E-protein from the same dimer (Fig. 2a, Supplementary Fig. 3).

Domain III of TBEV E-protein has an immunoglobulin-like fold (Fig. 2c). Antibodies targeting domain III are more likely to neutralize the virus than those interacting with other parts of the ectodomain[32]. There is evidence that neutralizing antibodies binding to domain III prevent pH-induced conformational changes of E-proteins, which are required for membrane fusion, or sterically block receptor binding[27,33].

The structures of E-protein ectodomains within the TBEV virion and of the isolated E-domain solved previously by X-ray crystallography[27] have an root-mean-square deviation (RMSD) of

1.7 Å for the corresponding Cα atoms. The most important difference is in the positioning of domains I–III relative to each other. Whereas in the crystal structure the domains I, II, and III are arranged in a line, in the virion the tip of domain II is bent 15 Å toward the virus membrane (Fig. 2c). A similar hinge-like movement of the domain II was described previously for E-proteins of WNV and DENV[8,28,34]. The bending of the ectodomain in the virion is necessary to keep the fusion loop buried in the hydrophobic pocket of the other E-protein from the same dimer, so that the loop is prevented from untimely induction of membrane fusion.

The C-terminal domain IV anchors the E-protein in the virus membrane. Domain IV is composed of five helices (Fig. 2c). The three N-terminal helices are perimembrane, whereas the last two are transmembrane. The perimembrane helices located in the outer leaflet of the virus membrane are amphipathic. The

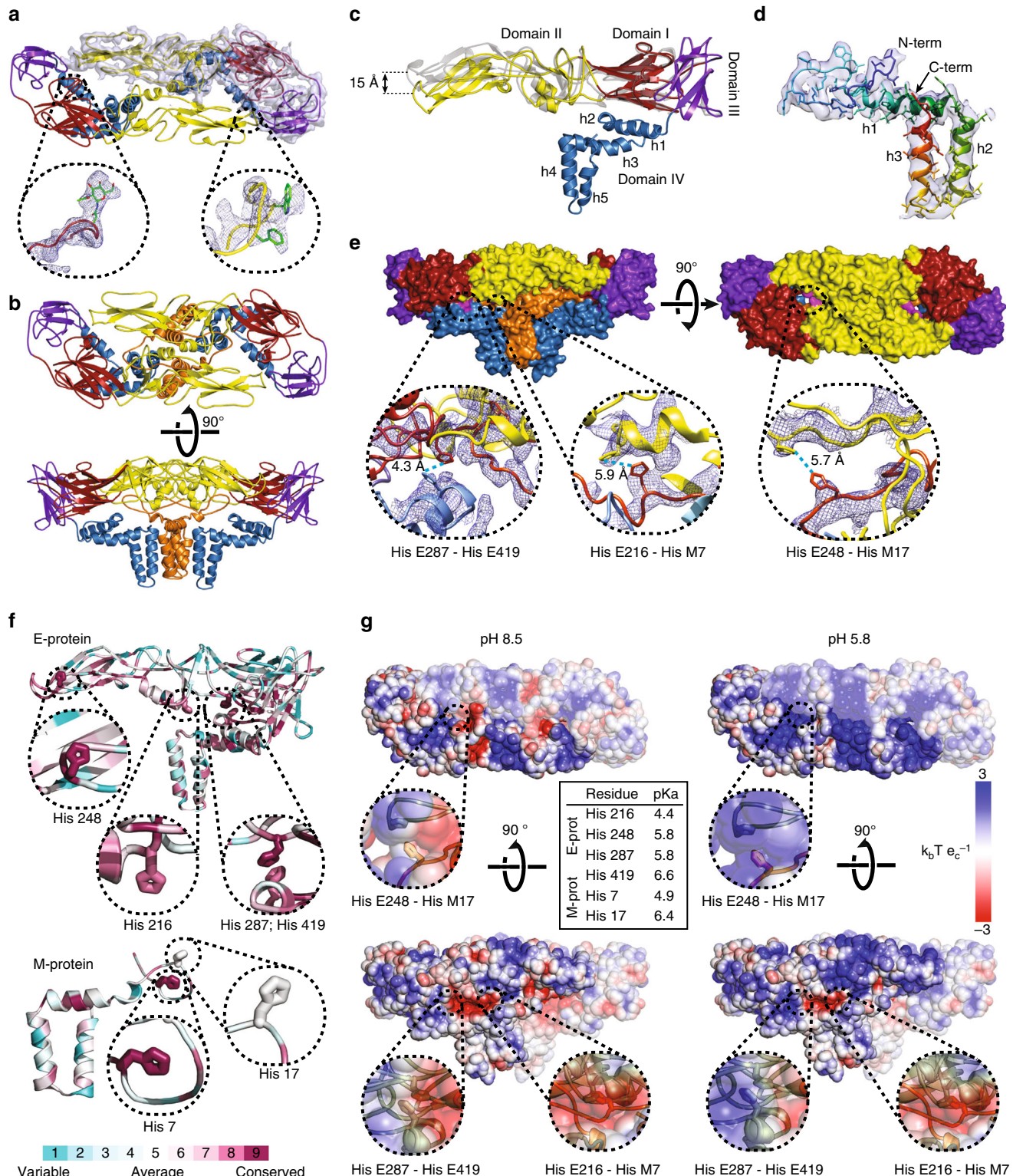

transmembrane helices are mostly hydrophobic. In the cryo-EM reconstruction of the TBEV virion, the domains IV are less well resolved than the ectodomains (Fig. 1d, Supplementary Fig. 1c). The virus membranes are acquired during budding, and are likely to be variable in lipid composition and to some extent also in the number of lipid molecules that are present in each virion. The variations in the virus membranes are likely to affect the positioning of E-protein helices in particular virions. Therefore, averaging the images of many particles during the process of three-dimensional reconstruction results in a smearing of details of the helices interacting with the membrane. In contrast, the ectodomains are tightly packed on the virus surface and their relative movements are limited.

**Structure of the M-protein.** Due to its small size and association with the virus membrane, the M-protein is not exposed at the virion surface. Residues 2–72 out of 75 of the M-protein could be built in the cryo-EM electron density map of TBEV. The M-protein consists of an N-terminal loop and three helices (Fig. 2d). The first helix is perimembrane and the last two are transmembrane.

Two M-proteins and two E-proteins form a heterotetramer in which each M-protein interacts with both E-proteins (Fig. 2b, Supplementary Fig. 4). This complex is the basic building block of the mature virion. The N-terminal loop of the M-protein interacts with domain II of the E-protein and presumably prevents the reorganization of E-protein dimers into fusogenic trimers[8]. The membrane part of the same M-protein interacts with the transmembrane domain of the other E-protein from the same heterotetramer, thus stabilizing the E-protein dimer (Fig. 2b, Supplementary Fig. 4).

**Role of histidines in putative pH sensing mechanism.** Flaviviruses deliver their genomes into the cell cytoplasm by fusing the virus and endosome membranes[35]. This fusion is induced by trimers of E-proteins that form when the virions are exposed to low pH in the endosomes[11,33]. The environment in endosomes with pH lower than 5.8 can cause protonation of side chains of histidines, which become positively charged. It was proposed that in the dengue virus, the protonated His7 from the M-protein and His208 from the E-protein repel each other and induce the disruption of E–M heterodimers[8]. The E-proteins can then form trimers and induce membrane fusion[8]. TBEV homologs His7 and His216 are located 5.9 Å away from each other (Fig. 2e), and are therefore likely to have the same function in sensing low pH. Additionally, His17 of the TBEV M-protein and His248 of the E-protein are separated by 5.7 Å, and after protonation might also contribute to heterodimer destabilization (Fig. 2e). His7 of the M-protein is conserved among many flaviviruses, whereas His17 is

only conserved among tick-borne flaviviruses (Fig. 2f, Supplementary Fig. 5), indicating that there might be a unique mechanism of structure destabilization for this group of viruses. Furthermore, His287 and His419 of the TBEV E-protein are located 4.3 Å from each other and may electrostatically repel each other when protonated at low pH (Fig. 2e). His287 is part of domain I, whereas His419 belongs to the second perimembrane helix of the E-protein (Fig. 2c, e). Repulsion between these amino acids is likely to trigger the release of ectodomains of E-proteins from the virus envelope, enabling the formation of fusion trimers. Homologs of His287 and His419 of TBEV are present in several other flaviviruses, and the mechanism for inducing detachment of the E-protein ectodomain from the virus membrane might be shared within the virus family (Fig. 2f, Supplementary Fig. 5). Four out of the six histidines that interact with each other within the TBEV heterotetramer have pKa values equal to or higher than 5.8 (Fig. 2g). This provides additional evidence that the histidines could serve as pH sensors. These pKa calculations are sensitive to the precise locations of amino-acid side chains within the protein structure[36]. It is therefore important that the role of histidines in controlling the pH-mediated conformational switch of the flavivirus E-proteins is supported by previous experimental evidence. Nelson et al. demonstrated that WNV E-proteins with single mutations in histidines do not differ from the wild-type virus in their capacity to induce membrane fusion[37]. Similarly, Fritz et al. showed that single mutants of TBEV E-protein His248Asn and His287Ala could induce membrane fusion with an efficiency similar to that of the wild-type[38]. However, the double mutant His248Asn and His287Ala had a lower efficiency of formation of E-protein trimers, and its sub-viral particles were nearly fusion incompetent. In combination, the mutational and structural analyses provide evidence that the disruption of E–M heterodimers and detachment of the E-protein ectodomains from the virion membrane may depend on the protonation of histidines in the low pH of late endosomes.

Patches of the heterotetramer surface became positively charged at pH 5.8, which approximates that of the late endosomes (Fig. 2g). The alteration of the surface charge distribution may further contribute to the initiation of the conformational changes required for the formation of pre-fusion E-protein trimers.

**Structure of TBEV virion covered with Fab 19/1786.** Mouse monoclonal antibody IgG1 19/1786 has therapeutic potential because it neutralizes multiple strains of TBEV and has minimal cross-reactivity with other flaviviruses[39]. We determined the EC$_{50}$ values for the whole antibody and Fab fragment to be 0.24 ± 0.03 and 35.0 ± 2.5 µg ml$^{-1}$, respectively (Fig. 3a, b). It is common that the inhibiting concentration of Fab is more than 100 times higher than that of the full antibody[40]. After the incubation of TBEV

**Fig. 2** Structure and organization of the E-proteins and M-proteins in TBEV virion. **a** Dimer of E-proteins with domain I colored in red, domain II in yellow, domain III in violet, and domain IV in blue. The electron density map of one of the proteins is shown as a semi-transparent surface. Glycosylation site Asn157 and residues Trp101 and Phe108 from the fusion loop of domain II are shown in detail. **b** Heterotetramer of two E-proteins and two M-proteins. E-proteins are colored according to domains, and M-proteins are shown in orange. **c** Superposition of cryo-EM (colored) and X-ray (gray) E-protein structures[27]. The cryo-EM structure includes three perimembrane (h1–h3) and two transmembrane helices (h4 and h5). **d** M-protein rainbow-colored from N-terminus in blue to C-terminus in red with electron density map shown as semi-transparent surface. The M-protein consists of an extended N-terminal loop followed by perimembrane (h1) and two transmembrane helices (h2 and h3). **e** Molecular surface of E–M heterotetramer. Histidines with a putative role in the dissociation of the heterotetramers are shown in magenta. Insets show details of interactions of the histidine side chains. **f** Structures of E-proteins and M-proteins colored according to conservation of amino acid sequence among viruses from the family *Flaviviridae* (for more details, see Supplementary Fig. 5). Insets show the conservation of histidines that might be involved in the pH-dependent dissociation of the heterotetramers. **g** Molecular surface of E–M heterotetramer colored according to electrostatic potential at pH 8.5 and 5.8. One E–M heterodimer is shaded for clarity. Insets show the surface potential surrounding the histidines that might be involved in the pH-dependent dissociation of the hetrotetramer. The table in the middle lists the pKa values of the selected histidines calculated using Rosetta-pKa[36]

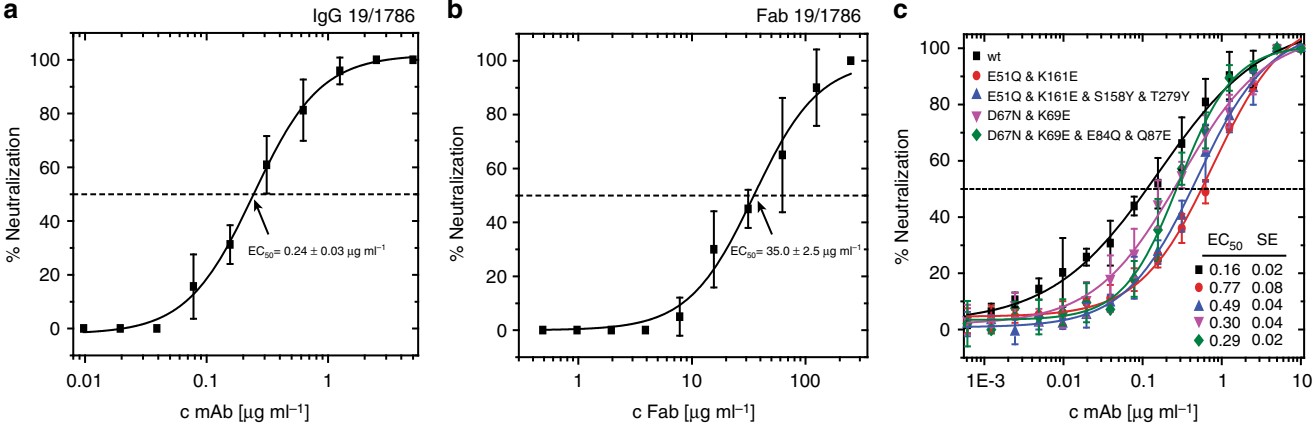

**Fig. 3** Neutralization of TBEV by antibody 19/1786. Dose-dependent neutralization curves of TBEV treated with IgG 19/1786 (**a**) and Fab 19/1786 (**b**). Error bars represent standard deviation of the measurements (IgG $n = 5$; Fab $n = 2$). $EC_{50}$ values with standard error of the mean are shown in the graph. **c** Dose-dependent neutralization curves of TBEV and its mutants treated with IgG 19/1786. Mutations in domain I (red circle—double mutant; blue triangle—quadruple mutant) and domain II (magenta triangle—double mutant; green diamond—quadruple mutant) affect the neutralization activity of Fab 19/1786 compared to the wild-type virus (black square). The mutations in the domains are listed in the top left corner. The error bars represent the standard deviation of the measurements ($n = 3$). Calculated $EC_{50}$ values ($\mu g\,ml^{-1}$) with standard errors are listed in the bottom right corner

with Fab fragments of 19/1786 the virions became "spiky" in appearance, confirming that the Fab fragments had attached to the virus (Fig. 4a). Cryo-EM reconstruction of the TBEV–Fab complex was determined to a resolution of 3.9 Å (Fig. 4b, Supplementary Fig. 6, Table 1). Two Fab fragments attached to each icosahedral asymmetric unit of the TBEV virion (Fig. 4b, c, d). The binding of the Fab fragments did not induce any major changes in the virion structure. The electron density map of the Fab fragments enabled the building of the structure of the variable loops of the antibodies that are responsible for virus recognition (Supplementary Figs. 7, 8). The constant parts of the Fab fragments distant from the virus surface were less well resolved, indicating the flexibility of the complex (Fig. 4b, Supplementary Fig. 6b).

Because of the non-quasi-equivalent organization of the TBEV particle, the two Fab fragments bound within one asymmetric unit differ in some interactions with the E-proteins. One of the Fab fragments binds next to the icosahedral 3-fold axis of the TBEV envelope (Fig. 4d). The Fab interacts with domain III of one E-protein and domain I of the E-protein from a neighboring asymmetric unit (Fig. 4d, Supplementary Fig. 7). The other Fab fragment binds close to the icosahedral 5-fold axis to domain III and interacts with domain II of an E-protein from a neighboring asymmetric unit (Fig. 4d, Supplementary Fig. 8). The Fab 19/1786 could not bind to the third E-protein within the icosahedral asymmetric unit because upon binding to domain III of the unoccupied E-protein, the heavy chain of the Fab would clash with domain III of a neighboring E-protein (Supplementary Fig. 9). Therefore, each TBEV virion can bind up to 120 Fab fragments of antibody 19/1786. All the Fab attachment sites can be occupied by complete antibodies of type G without steric hindrance (Supplementary Fig. 10). It is notable that antibodies E16 and ZV-54/ZV-67, which can neutralize WNV[41] and ZIKV[42], respectively, also interact with the domain IIIs of E-proteins and bind to the virus particles with the same stoichiometry as that of 19/1786. The Fab fragments of antibody 19/1786 were mixed with the virus in an equimolar ratio relative to E-proteins. However, the electron density map of the complex had lower-density values in the regions corresponding to Fab than in the regions of the viral envelope. This indicates that the Fab fragments did not have full occupancy. Using a localized 3D classification technique[43], the occupancy of the Fab fragment

bound close to the 3-fold icosahedral axis was determined to be 70%, whereas that of the fragment bound close to the 5-fold axis was 60%.

The major interaction site of the 19/1786 Fab fragment with the virus is the domain III of the E-protein. The 19/1786 Fab fragments bind to this site at an angle of 135° from the virus surface (Fig. 4e). The same amino acids of hypervariable regions of heavy and light chains of the antibody are in contact with the amino acids of the domain III at both of the attachment sites within the icosahedral asymmetric unit. The total buried surface area of the interface is 730 Å$^2$. The interaction is formed mainly via salt bridges and hydrogen bonds. Part of the heavy chain of the Fab bound close to the 3-fold axis interacts with domain I of the E-protein through an interface with a buried surface area of 300 Å$^2$ (Supplementary Fig. 7). Part of the Fab heavy chain interacts with the domain II close to the 5-fold axis through a buried surface area of 370 Å$^2$ (Supplementary Fig. 8). One of the E-proteins from the icosahedral asymmetric unit interacts with three Fabs, the second E-protein with one Fab and the third E-protein does not interact with any (Fig. 4d).

It was shown previously that the replacement of Thr310 in the TBEV E-protein with another amino acid resulted in a decreased infectivity of the mutant virus[44]. It was speculated that this residue is important for receptor recognition[44]. Thr310 is part of the 19/1786 binding site (Supplementary Figs. 7, 8). It is therefore possible that the interaction of antibody 19/1786 with domain III may interfere with the binding of TBEV to its putative receptor.

To determine whether the interaction of antibody 19/1786 with domains I and II had any role in the virus neutralization, we prepared viruses with mutations in the antibody binding sites. In domain I two amino acids that form hydrogen bonds with the heavy chain of the antibody (Supplementary Fig. 7) were mutated to amino acids with opposite charges (double mutant Glu51Gln and Lys161Asn). To further disrupt the interaction interface, a quadruple mutant was prepared in which two additional amino acids with small side chains were replaced with tyrosine (Glu51Gln, Lys161Asn, Ser158Tyr, and Thr279Tyr). In order to disrupt the binding site of the antibody in domain II, two and four amino acids were replaced with residues with opposite charges (double mutant Asp67Asn, Lys69Glu and quadruple mutant Asp67Asn, Lys69Glu, Glu84Gln, and Gln87Glu). The mutations in its domain I made the virus more resistant to

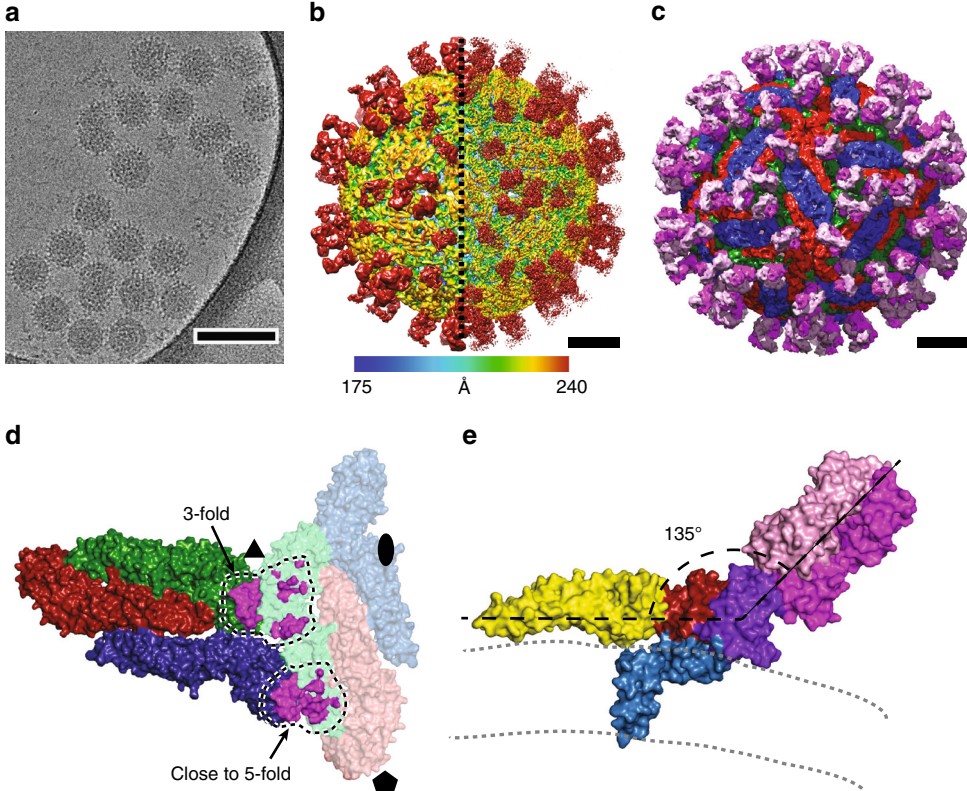

**Fig. 4** Interaction of TBEV virions with Fab fragments of neutralizing antibody 19/1786. **a** Cryo-EM micrograph of TBEV virions incubated with Fab fragments of 19/1786. Scale bar represents 100 nm. **b** Electron-density map of Fab-covered TBEV virion rainbow-colored according to distance from center of particle. The right half of the image represents a B-factor sharpened map. Electron densities corresponding to the Fab fragments are located close to the 3-fold and 5-fold symmetry axes of the virion. **c** Molecular surface of TBEV virion covered with Fab 19/1786 fragments low-pass filtered to 7 Å resolution. E-proteins are shown in red, green, and blue. Fab fragments are shown in magenta (heavy chain) and pink (light chain). Scale bars in **b** and **c** represent 10 nm. **d** Footprints of Fab 19/1786 on TBEV surface. Fab fragments bind to two of the E-proteins of the asymmetric unit. The Fabs interact with an additional E-protein from the neighboring asymmetric unit shown in faint colors. **e** The Fab 19/1786 binds to the domain III at an angle of 135° relative to the axis of the E-protein ectodomain. The E-protein domain I is shown in red, domain II in yellow, domain III in violet, and domain IV in blue. The heavy chain of the Fab fragment is shown in magenta and the light chain in pink. The leaflets of the viral membrane are represented by gray dashed lines

antibody 19/1786 neutralization, with $EC_{50}$ values increasing from 0.16 µg ml$^{-1}$ for the wild-type virus to 0.77 µg ml$^{-1}$ and 0.49 µg ml$^{-1}$ for the double and quadruple mutants, respectively ($p$-values < 0.0001; Fig. 3c). The quadruple mutant exhibited lower resistance to the neutralization than the double mutant ($p = 0.0068$). Presumably the additional mutations in its domain I served as suppressor mutations instead of having the expected synergic effect. A less pronounced reduction in the neutralization activity was observed for the mutants of domain II, with $EC_{50}$ values increasing to 0.30 and 0.29 µg ml$^{-1}$ for the double and quadruple mutants, respectively ($p$-values ≤ 0.0086, Fig. 3c). The difference in the $EC_{50}$ values of the two mutants in domain II is not statistically significant ($p = 0.835$). The results of these mutational experiments indicate that the interaction of antibody 19/1786 with domains I and II contributes to virus neutralization. The binding of the antibody to domain II may prevent the induction of membrane fusion, whereas the interaction with domain I may interfere with the hinge movement that is required for the formation of pre-fusion trimers[8]. Nevertheless, the interactions of antibody 19/1786 with domains I and II appear to be only auxiliary, and the function of the antibody probably depends mostly on its interaction with domain III.

**Fab 19/1786 may prevent TBEV membrane fusion.** Flaviviruses enter their host cells by receptor-mediated endocytosis[45]. The low pH in endosomes triggers a conformational rearrangement of the E-proteins that involves the formation of a cone-shaped trimer from E-protein ectodomains, which has fusion loops exposed at its tip[46]. The fusion loops interact with the endosome and trigger fusion of the endosome and virus membranes, resulting in the subsequent release of the virus nucleocapsid core into the cell cytoplasm[10,47].

Purified flavivirus particles exposed to a low-pH solution in vitro fuse with each other (Fig. 5a)[48]. The binding of Fab fragments of antibody 19/1786 prevented the fusion of TBEV virions at pH 5.8 (Fig. 5a). However, this might be caused by the inaccessibility of the virus membrane at the surface of the Fab-decorated TBEV virions. To determine whether IgG 19/1786 and Fab 19/1786 can prevent membrane fusion in vivo, we performed a "fusion-from-without" assay using C6/36 cells[49]. Whereas the native TBEV induces cell fusion at low pH, the virus in complex with IgG 19/1786 lost this ability, and the virus in complex with the Fab 19/1786 induced cell fusion with lower efficiency than the native virus (Fig. 5b). The incomplete inhibition of the fusion by the Fab fragments is in agreement with the results of the neutralization test, which show that the $EC_{50}$ value of the Fab is 150× higher than that of the full IgG (Fig. 3a, b).

Cryo-EM reconstruction of the TBEV–Fab complex at low pH with imposed icosahedral symmetry produced maps with a resolution limited to 19.2 Å, indicating that the particles are pleiomorphic (Fig. 5c, Supplementary Fig. 6a). The low-resolution map shows that the particles have lost the native

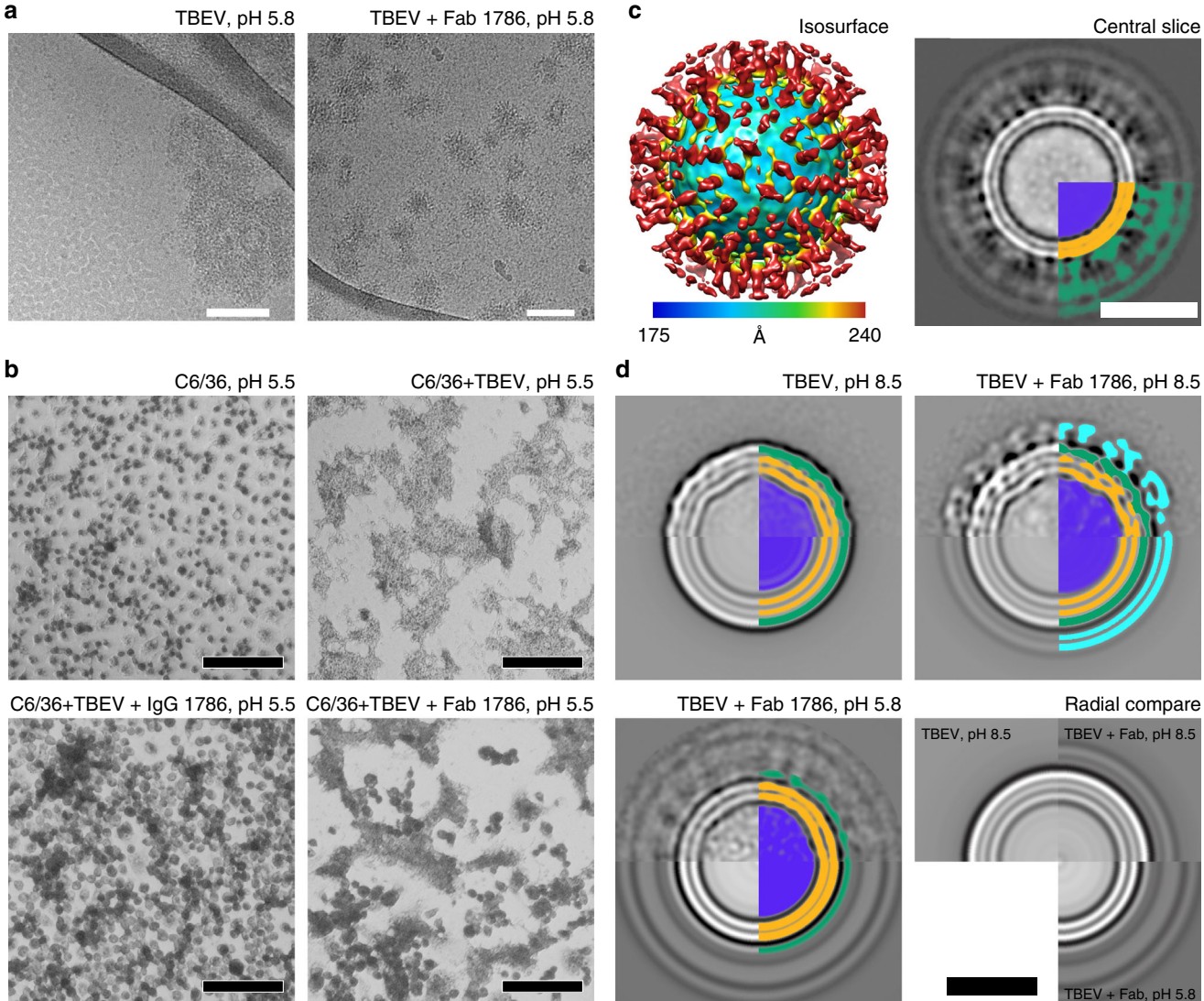

**Fig. 5** TBEV virions and TBEV–Fab 19/1786 complex in low-pH environment. **a** Cryo-EM micrographs of TBEV virions fused together under low-pH conditions. However, TBEV virions covered with Fab 19/1786 fragments did not aggregate. Scale bar represents 100 nm. **b** "Fusion-from-without" assay on C6/36 cells. The control cells did not fuse in the low-pH (5.5) environment. In contrast, native TBEV induced complete cell fusion. Pre-incubation of TBEV with IgG 19/1786 completely abolished the fusion activity of the virus and the Fab 19/1786 lowered the fusion activity of the TBEV. The scale bar represents 100 μm. **c** Cryo-EM reconstruction of Fab-covered TBEV virion at low pH. The isosurface is radially colored according to the distance from the center. Corresponding central slice perpendicular to the 5-fold axis is shown on the right. The particles lost the compact layer of the E-protein ectodomain and the viral membrane is not deformed by the transmembrane helices. The lower right quadrant of the slice is color-coded as follows: nucleocapsid—blue; inner and outer membrane leaflets—orange; M-proteins, E-proteins, and Fab fragments cannot be distinguished and are shown in green. The scale bar represents 25 nm. **d** Reference-free 2D class averages of TBEV virions in solutions with different pH levels. The upper half of the images shows the class averages, whereas the bottom parts are radial averages of the same classes. In the radial averages, the three distinct layers represent the inner and outer leaflets of the membrane (in orange) and the ectodomain of the E-proteins (in green). An additional double layer is visible on the virion covered with Fab fragments (in cyan). The layer corresponding to ectodomains of E-proteins and Fab fragments is more diffuse in the low-pH structure than in the particles at neutral pH. The viral membrane remained fully preserved, but lost the deformities introduced by the transmembrane helices of virus proteins. The 2D class averages are color-coded as follows: nucleocapsid—blue; inner and outer membrane leaflets—orange; E-proteins—green; Fab 19/1786 attached to virus surface—cyan. The scale bar represents 25 nm

organization of the E-protein ectodomains (Fig. 5c). This is corroborated by 2D class-averages of the virions (Fig. 5d). Particles with bound Fab fragments at low pH lack the density corresponding to the ectodomain layer, whereas the lipid bilayer enclosing the nucleocapsid core is intact (Fig. 5c, d). Notably, the leaflets of the lipid layer are spherical and have lost the deformations present in the native particles (Fig. 5c, d). This indicates a reorganization of the positions of transmembrane helices of the E-proteins and M-proteins. Most likely the proteins lost their icosahedral ordering and became irregularly distributed in the virus membrane. Even though the ectodomains of E-proteins detached from the virus membrane, the fusion capability of the virus became impaired because of the Fab binding. Chao et al. had shown that availability of competent monomers within the contact zone between virus and target membrane makes trimerization a bottleneck in hemifusion[11]. It is therefore possible that the Fab 19/1786 binding interferes with the conformational rearrangement of the E-protein dimers into fusogenic trimers.

The mechanisms of virus neutralization differ depending on the virus and the neutralizing antibody. Zika virus covered with Fab fragments of the C10 antibody exhibited extraordinary stability at low pH because the ectodomains of the E-proteins were locked in dimers similar to those in the native virus[50]. The crystal structure of a human antibody which is active against all DENV serotypes in complex with the E-protein ectodomain revealed an "E-dimer-dependent epitope" that includes the conserved main chain of the fusion loop and the two conserved glycosylation sites of the virus[51]. The Fab fragment of antibody 5J7 bound across all three E-proteins of one icosahedral asymmetric unit of DENV3 and neutralized the virus by a combination of locking the ectodomains in place and steric obstruction of the receptor binding[30]. Steric hindrance of the conformational rearrangement of E-proteins is also the proposed neutralization pathway of the E16 Fab complex with WNV[52]. A neutralization mechanism similar to that of Fab 19/1786 was observed for DV2-E104 Fab. The Fab fragments did not lock the DENV2 E-protein in dimers; however, they inhibited the membrane fusion process[53]. Because the IgG 19/1786 antibody is not cross-reactive against other flaviviruses and efficiently neutralizes TBEV[39], it has potential for therapeutic use.

## Methods

**Production and isolation of mature TBEV particle.** Purified TBEV virions were prepared using a modified protocol for the dengue virus[6]. Human neuroblastoma cells UKF-NB4 were grown to 100% confluence in IMDM medium supplemented with 10% FBS at 37 °C in the presence of 5% $CO_2$ in 30 flasks, each with a bottom surface area of 300 $cm^2$. The cells were infected with the TBEV strain Hypr (low-passage TBEV strain isolated in 1953 from the blood of a deceased child with TBE) at an MOI of 0.5. After 5 h of incubation at 37 °C, the medium was replaced with fresh medium without FBS. The culture media were harvested 35 h post infection and clarified by centrifugation at $5700 \times g$ for 10 min at 4 °C. The supernatant was precipitated by adding PEG 8000 to a final concentration of 8% (w/v) and incubating overnight at 4 °C with mild shaking. After that, the virus was pelleted by centrifugation at $10,500 \times g$ for 50 min at 4 °C. The resulting pellet was resuspended in 2 ml of NTE buffer (20 mM Tris, 120 mM NaCl, 1 mM EDTA, pH 8.5). The solution was clarified by centrifugation at $1500 \times g$ for 5 min at 4 °C. RNAse was added to the supernatant to a final concentration of 10 μg $ml^{-1}$ and incubated for 15 min at 10 °C. The solution was loaded onto a step tartrate gradient (10, 15, 20, 25, 30, and 35%) in NTE buffer. After separation in a Himac CP80WX ultracentrifuge (Hitachi) with a P40ST swinging bucket rotor at 32,000 rpm for 2 h at 4 °C, a visible band containing the virus was harvested using a syringe with a needle. Finally, the collected virus was repeatedly diluted with 4 ml of NTE buffer and concentrated to a final volume of 100 μl using a centrifugal filter concentrator with a 100-kDa cut off (Vivaspin® 6 Centrifugal Concentrator, Vivaproducts).

**Virus neutralization assay.** Virus neutralization by the 19/1786 mAb and corresponding Fab fragments was measured according to a previously published protocol[39]. Briefly, serial dilutions of the antibody and Fab fragment were prepared, mixed with TBEV (1000 PFU $ml^{-1}$), and incubated at 37 °C for 2 h. After the incubation, the mixtures were applied to monolayers of porcine kidney stable cells in 96-well plates, and incubated for 4 days at 37 °C. The cytolysis was examined using light microscopy and the neutralization rate was determined. The assay was done in triplicates for the IgG and in duplicates for the Fab. Neutralization curves were constructed from the measured data and $EC_{50}$ values with corresponding standard errors were calculated from the fitted Hill dose-response curve.

**C6/36 cell fusion assay.** A fusion-from-without assay was performed as described previously[49]. Mosquito C6/36 cells were grown in 96-well tissue cell culture plates for 2 days. The cells were precooled for 45 min at 4 °C, then washed with serum-free medium. Cells were incubated for 1 h at 4 °C with 30 μl of purified virus at a concentration of 500 μg $ml^{-1}$ or a mixture of virus pre-incubated (30 min) with IgG 19/1786 (100 μg $ml^{-1}$) or Fab 19/1786 (3000 μg $ml^{-1}$). After removal of the virus suspension, pre-warmed fusion medium (MEM buffered with 20 mM MES, pH 5.5) was added to the cells and the plates were incubated for 2 min at 40 °C. Fusion medium was replaced with a growth medium, the cells were further incubated at 40 °C for 2 h, and then the cells were fixed with a 1:1 mixture of methanol and acetone and stained with Giemsa's solution.

**Preparation of E-protein mutants and neutralization assay.** Recombinant TBEV (Oshima 5–10 strain) was prepared from infectious cDNA clones, Oshima-IC as reported previously[54]. To introduce mutations, cDNA fragments with the mutations were synthesized by standard fusion-PCR and subcloned into Oshima-

IC in a stepwise manner. Infectious RNA was transcribed from Oshima-IC using mMESSAGE mMACHINE SP6 (Thermo Fisher) and transfected into BHK-21 cells using TransIT-mRNA (Mirus Bio LLC), as described previously[54]. Recombinant viruses were recovered from cell culture supernatants.

To measure the ability of antibody 19/1786 to neutralize the TBEV, the mutant viruses were incubated with serially diluted antibody and inoculated into BHK-21 cells. The cells were grown in minimal essential medium containing 1.5% carboxymethyl cellulose and 2% FBS for 4 days. After 4 days of incubation, the cells were fixed with 10% formalin and stained with 0.1% crystal violet. Plaques were counted and expressed as plaque-forming units (PFU $ml^{-1}$), and the reduction in the number of plaques by the antibody was evaluated. The neutralization experiments were done in triplicates. Neutralization curves were constructed from the measured data and $EC_{50}$ values with corresponding standard errors were calculated from the fitted Hill dose-response curve. For statistical comparison of the means of groups, an unpaired $t$-test was used and the corresponding $p$-values are reported.

**Preparation and sequencing of Fab 19/1786.** Fab 19/1786 fragments were prepared and purified with a Pierce™ Fab Preparation Kit (Thermo Scientific) from mouse monoclonal IgG1 19/1786[39]. The amino acid sequence of the variable regions of the antibody was determined by mass-spectroscopy analysis and by sequencing cDNA from hybridoma cells according to Wang et al[55].

**Cryo-EM sample preparation and data acquisition.** To prepare the virus–Fab 19/1786 complex, TBEV particles were incubated with the Fab 19/1786 for 2 h at 4 °C, using equimolar amounts of the Fab fragments and E-proteins. To study the mechanism of virus neutralization by Fab 19/1786, the pH of the sample was adjusted to 5.8 with 100 mM MES pH 5.5 and the sample was incubated for 15 min at 4 °C. Samples for cryo-EM were vitrified using an FEI Vitrobot Mark IV on Quantifoil R2/1 grids with the following settings: 3.8 μl sample; wait time 10 s; blot time 2 s; blot force −2.

The grids with vitrified virions were loaded into an FEI Titan Krios microscope operating at 300 kV, equipped with an FEI Falcon II direct electron detector. The microscope illumination and projection system was aligned before data acquisition, and the astigmatism and coma-free alignments were corrected every 12 h during the acquisition process. The micrographs were acquired using the automated acquisition software EPU (FEI) at defoci varying between 1 and 3 μm at 75,000× magnification, resulting in a pixel size of 1.063 Å. Six acquisition areas were defined per foil-hole and autofocus was performed before the acquisition of each foil-hole. Images were recorded as seven-frame movies, with a total exposure time of 0.5 s and dose of 22 $e^-$ $Å^{-2}$.

**Data processing and volume reconstruction.** The seven-frame movies were aligned and summed using the program motioncor2[56]. The contrast transfer function (CTF) of the micrographs was estimated by gCTF[57]. Due to the high heterogeneity of the sample, the particles were manually boxed using e2boxer from the package EMAN2[58]. Because of the large resulting box size, the particles were down-sampled using XMIPP FFT binning[59] to a box size of 512 × 512 pixels, which resulted in a pixel size of 1.46 Å.

Particle images of native TBEV (19,111) and TBEV–Fab complex (12,098) were subjected to several rounds of 2D and 3D classification performed using the software package RELION[60]. An electron density map of Dengue virus 2 (EMDB: EMD-5520)[8] low-pass filtered to 60 Å was used as an initial model for 3D classifications and refinements of TBEV. The classification steps resulted in the selection of 11,882 particles of native TBEV and 5929 particles of the TBEV–Fab complex, which were used for the final reconstruction according to the gold standard using the 3Dautorefine procedure in RELION. The resulting maps were masked and B-factor sharpened using the post-process procedure in RELION[61]. Resolutions of the reconstructions were determined as points where FSC fell below 0.143. Local resolutions of the maps were determined using the post-process procedure in RELION.

For the reconstruction of the TBEV–Fab complex at low pH, single non-overlapping particles were boxed from the micrographs. Reference-free 2D classification was used to remove damaged particles. The TBEV structure low-pass filtered to 60 Å was used as an initial model. The particles were subjected to 3D classification; however, this approach did not lead to a single class of uniform particles, but instead in each iteration the particles redistributed randomly among the three generated classes. Thus, all the particles that passed the 2D classification were used for the reconstruction process, which did not lead to a high-resolution map. We repeated the reconstruction with C1, C5, and icosahedral symmetries, as well as with masks of different sizes in an attempt to remove the most pleiomorphic parts of the particles from the orientation determination process. The best results were achieved by masking out the region including the Fab fragments, and aligning the particles only according to the features of the underlying TBEV particle. The data set was homogenized by 3D classification that used the orientations of the particles from the previous reconstruction. This approach partially eliminated some of the variability in the region containing the Fab fragments. The final reconstruction based on 3831 particle images was calculated using RELION.

**Sub-particle reconstruction to determine Fab occupancy**. To quantify the occupancy of Fab fragments at the two E-protein interaction sites, sub-particles of Fab fragments were extracted from the particle images using the localized reconstruction tools in RELION[43]. Orientations of the virus particles determined during the 3D reconstruction were used for the extraction process. The sub-particles were 3D-classified using RELION, and the number of particles in classes representing occupied and unoccupied sites were summed. Because the Fab fragments present at the 3-fold axis of the particle were too close to each other to extract separately, they were extracted as one sub-particle and a tight mask around each Fab fragment was used during the 3D classification process.

**Model building**. Cryo-EM maps of native TBEV and the TBEV–Fab complex were re-oriented so that the 222 subset of icosahedral symmetry axes was aligned with the Cartesian coordinate axes. The maps were cropped, normalized, and set to crystallographic P23 symmetry. This treatment of the maps resulted in five icosahedral asymmetric units in one "crystallographic" asymmetric unit of the P23 space group and enabled efficient refinement of the structures.

An initial model of the E-protein was generated based on the known crystal structure of the ectodomain (PDB:1SVB)[27] and the transmembrane domains of Dengue virus type 2 and the Zika virus (PDB:5IRE,3J27)[8,23] using the program Modeller[62]. The model was rigid-body fitted to the electron density map of the TBEV particle using the program Chimera[63]. Subsequently, the structure was manually corrected using the program Coot[64], followed by real-space refinement in Phenix[65] and reciprocal space refinement in Refmac5[66]. The model of the M-protein was built in the same manner. The complete icosahedral asymmetric unit containing three E-proteins and three M-proteins was refined using the program Refmac5 with imposed icosahedral symmetry constraints.

The initial homology model of the Fab 19/1786 was generated using the program Modeller based on the PDB structures of multiple Fabs (PDB: 2H1P, 3FFD, 3LIZ, 5DO2, 5T6P, 4AEI, 5B4M, 1MEX). The resulting model was rigid-body fitted to the measured electron density and refined together with the interacting E-proteins using the real-space refinement procedure in Phenix[65].

**Analysis of the molecular models**. The quality of the structures was assessed with the MolProbity server[67] and wwPDB validation service. Intermolecular interfaces and interacting amino acids were identified using the programs PDBePISA and UCSF Chimera[63]. Protein sequences were compared using the program Clustal Omega, and the conservation of the amino acids was visualized using the program Consurf[68]. Predictions of pKa were calculated using Rosetta-pKa[36], according to the method described for DENV E-protein by Chaudhury et al[69]. Electrostatic surfaces were visualized in the program PyMOL using the APBS and PDB2PQR plugins.

**Data availability**. Cryo-EM electron density maps of the native TBEV virion and its complexes with Fab fragments at neutral and low pH have been deposited in the Electron Microscopy Data Bank, https://www.ebi.ac.uk/pdbe/emdb/ (accession numbers EMD-3752, EMD-3754, and EMD-3755), and the fitted coordinates have been deposited in the Protein Data Bank, www.pdb.org (PDB ID codes 5O6A and 5O6V, respectively). The additional data that support the findings of this study are available from the corresponding author upon request.

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

## Acknowledgements

We wish to thank the Central European Institute of Technology Core Facilities Cryo-Electron Microscopy and Tomography and Proteomics, supported by the Czech Infrastructure for Integrative Structural Biology (LM2015043 funded by the Ministry of Education, Youth and Sports of the Czech Republic) for their assistance in obtaining the scientific data presented in this paper. This research was carried out under the project CEITEC 2020 (LQ1601), with financial support from the Ministry of Education, Youth and Sports of the Czech Republic under National Sustainability Program II. Computational resources were provided by the CESNET LM2015042 and the CERIT Scientific Cloud LM2015085, provided under the programme "Projects of Large Research, Development, and Innovations Infrastructures". This work was supported by The Ministry of Education, Youth and Sports from the Large Infrastructures for Research, Experimental Development and Innovations project "IT4Innovations National Supercomputing Center − LM2015070". The Titan Xp used for this research was donated by the NVIDIA Corporation. The research leading to these results received funding from the European Research Council under the European Union's Seventh Framework Program Grant (FP/2007-2013)/ERC Grant Agreement 355855 (to P.P.), EMBO Grant Agreement IG 3041 (to P.P.), Czech Science Foundation Grant No. 17-02196S (to P.P. and D.R.), and JSPS KAKENHI Grant Numbers 16K1503206 and 17H03910 (to K.Y.).

## Author contributions

P.P. and D.R. designed research; T.F., P.F., and K.Y. performed research; M.N. contributed new reagents/analytic tools; T.F. and P.P. analyzed data; and T.F. and P.P. wrote the paper.

## Additional information

**Competing interests:** The authors declare no competing financial interests.

