## [Peer Review File · Nature Communications]

Reviewers' comments:

Reviewer #1 (Remarks to the Author):

This study presents two structures of TBEV virions with and without a neutralising Fab. It proposes that additional histidine residues contribute to an extended pH-switch compared to other flaviviruses. It also presents a mechanism of action of the neutralising antibody based on the structural results and imaging at low pH.

The results are technically sound and will be of interest to the flavivirus scientific community. Given the availability of several of similar quality for several other Flaviviridae (e.g. Zika, 3.7Å; DENV, 3.6Å, JEV, 4.3Å), the main interest for the broader virology community is the mechanism of the Fab inhibition. Indeed, high-resolution structure of virion-Fab complexes are more unusual (e.g. DEN2 and 3, 6.5Å and 9Å; Zika, 4-12Å). Unfortunately, the study does not clearly elucidate the mechanism of neutralization, showing some indication of inhibition of both receptor-binding and fusion.

The mechanistic aspects are less strongly supported than the first part of the manuscript. In particular, two points require additional data or analyses:

1. Two additional pairs of histidine residues are proposed to contribute to the pH-sensitive switch. While somewhat supported by the inter-histidine distance, one would expect more detailed analyses (pKa calculations; modelling) if not mutagenesis. Support from The previously published mutagenesis is tenuous.
2. Due to pleiomorphism, the acidic pH reconstruction has a low resolution and is possibly unreliable. Validation is again needed here to understand at what stage to the fusion might be blocked (e.g. compared to stages described in Chao et al. eLife 2014;3:e04389).

Overall, the manuscript presents useful structural details of the TBEV organisation with and without a neutralising Fab. However, in the absence of experimental data directly addressing the mechanism of pH-induced reorganisation and Fab inhibition, the advances may seem incremental outside of the flavivirus field compared to results established for DENV, WNV or Zika virus.

Further comments:

1. It is surprising that the following two articles are not discussed:
 - Rouvinski A et al. Nature 2015;520(7545):109–13.
 - Zhao H et al. Cell 2016;166(4):1016–27.
2. At this resolution, outliers in the Ramachandran plot need to be corrected or their validity discussed.
3. The resolution of the constant domain of the Fab is surprisingly low. Have lower symmetry reconstructions been attempted? Also, there is no mention of movie-refinement or particle polishing. This is likely to further improve the quality of the resulting maps.
4. A 3-D representation of sequence conservation (e.g. Consurf) around the histidine residues discussed in the text would be useful to complement Figure S3.

Reviewer #2 (Remarks to the Author):

In this study, Fuzik et al. report the cryo-EM structure of the TBEV virion alone and when bound by the monoclonal antibody 19/1786. While prior publications have reported the crystal structure of the TBEV E protein, and the cryo-EM structure of a TBEV subviral particle, this represents the first structure of the TBEV virion. As expected, the virion structure was found to be very similar to other solved flavivirus virion structures (WNV, DENV, etc.). A comparison of the E proteins in the cryo-EM structure to the previously determined crystal structure elucidated a bend in the protein that reflects its orientation in the context of the spherical virion, as opposed to a monomer in solution. A detailed analysis of the binding characteristics of the monoclonal antibody 19/1786 supports binding of this antibody at the 3- and 5-fold axes of symmetry, but not the 2-fold. A particularly novel aspect of this study is the finding that the antibody footprint binds distinct domains at each of these sites. While the epitope is primarily focused on E domain III, when bound near the 3-fold symmetry axis the antibody also contacts residues in domain I, and when bound near the 5-fold symmetry axis the additional contacts are found in domain II. Mutagenesis studies could be particularly interesting- can mutations in the antibody footprint of domain I or II inhibit binding of the antibody at one symmetry axis, but not the other? Is only one of these binding footprints responsible for inhibiting fusion? Overall, the manuscript is very well written, and the figures are clear. One piece of data that should be included is more information on the neutralizing activity of antibody 19/1786, including a neutralization curve/ experiment, as well as the EC50 concentration. The reference to the antibody is a paper from 1994 that is not easily accessible to readers, even those with broad journal access.

Additional comments include:

Lines 31-34. The authors state in the abstract “We show that the repulsive interactions of histidine side chains, which are protonated at low pH, disrupt heterodimers of TBEV envelope and membrane proteins and induce detachment of the E protein ectodomains from the virus membrane.” However, this is not experimentally shown, only hypothesized based on the proximity of specific histidine residues in the virus structure. Furthermore, a prior study in WNV did not support the histidine switch hypothesis (PMID: 19776132). The wording should be adjusted to not overstate the findings.

Lines 120-122. “The E-proteins of TBEV, WNV, ZIKV, and Japanese encephalitis virus contain a single homologous glycosylation site, whereas, that of DENV has an additional glycosylation site at Asn67.” While true that the majority of these virus strains are glycosylated, there are strains of WNV that are not glycosylated, and perhaps other viruses as well. I suggest changing this to say “most”, or “the majority”.

Supplementary Figure 3. Please indicate the strain of WNV (I assume NY99). The symbols denoting conservation versus absolute conservation are incorrect based on the sequence alignment. “*” should be absolute conservation, and “:” conservation. In the sequence alignment, WNV is written incorrectly as WNW.

Reviewer #3 (Remarks to the Author):

The cryo-EM structures are presented for the native TBEV virion and TBEV in complex with Fab fragments of neutralizing antibody 19/1786. The finding is novel because instead of locking the virus E-proteins into the native-like state, the authors believe the structure shows that repulsive interactions of histidine side chains disrupt heterodimers of TBEV envelope/membrane proteins to cause detachment of the E protein ectodomains from the virus membrane.

Minor issues: stylistic problems include:

- 1) Details in the abstract (repulsive interactions of histidine side chains) that might be better swapped with the lack of detail in the final paragraph of the intro (binding of Fab fragments does not prevent low-pH-induced movements of E proteins, but blocks membrane fusion - however, blocking membrane fusion is stated as conclusive, whereas it is suggestive).
- 2) Excessive use of short sentences leading to a choppy presentation, especially in introduction.

Line 100 - is the correct figure called here: "icosahedral asymmetric unit form unique

interactions with the surrounding glycoproteins (Fig. 1d)”

The authors are referring back to the central section of the map (Fig 1-d) to describe inner and outer leaflets, separation, shape, insertion, core shape etc without indicating any of these features in the figure. A color coded panel added to figure 2 might provide clarity and prevent the reader from shuffling back and forth between fig 1, 2, and 3 to follow the description.

Figure 1

Scale bars in d and e are same (10 nm), yet the surface rendered ‘d’ map is clearly larger than b or e.

Add arrow to b and e to indicate location of E and M-proteins.

The legend does not indicate if the map displayed as ‘b’ is a sharpened map

Figure 1 and 2

Keeping the capsid diameter the same between figures and panels would add clarity. Please address and define in the legend the difference between the rendering the maps used in Fig 1-a, 1-d, 2-b, 2-c, 3-a, 3-d. Some of these appear high resolution and some do not.

The local resolution map is difficult to interpret as rendered and the legend does not describe the local resolution map adequately. Is this a cut-away half map? Or is it a slabbed cross-section with noise shown in grey in the center?

Figure 4

The coloring in panel c is helpful to orient the viewer, but the figure would be improved if the maps in b and c were the same diameter and displayed adjacent to each other evenly. Again the change (between b and c) in contour or sharpening needs to be included in the legend.

Fig 6 is described by one line (220) and may be better suited to supplemental.

Line 243 is overstated, finding in the fab footprint a residue implicated by another work to be important to receptor binding does suggest the Fab binding might block receptor, but it is not as certain as stated.

Complex at low pH -- there were ~5000 particles in the 20Å map, but no other statistics were provided. The raw image shows overlapping capsids. There might be other contributing factors limiting resolution and the poor resolution of the map alone is not enough to allow some of the conclusions.

Line 270, overstatement that the ectodomains of E proteins detached from the virus membrane, but due to the Fab binding could not induce membrane fusion.

There is not enough evidence presented here to conclude absolutely that the Fab fragments inhibited the membrane fusion process. However, the comparisons presented in figure 7 are suggestive. This presentation would be helped by a color coded map (see above) to help the reader understand the three distinct layers representing which are the inner and outer leaflets of the membrane and the ectodomain of the E-proteins.

Insufficient details are provided for data collection in Methods.

Table 1 Ramachandran outliers 1.13% in virus map, is high

Response to reviewer's comments

Reviewer's comments are highlighted in blue italics, **our responses in bold black text.**

Reviewer #1 (Remarks to the Author):

This study presents two structures of TBEV virions with and without a neutralising Fab. It proposes that additional histidine residues contribute to an extended pH-switch compared to other flaviviruses. It also presents a mechanism of action of the neutralising antibody based on the structural results and imaging at low pH.

The results are technically sound and will be of interest to the flavivirus scientific community. Given the availability of several of similar quality for several other Flaviviridae (e.g. Zika, 3.7Å; DENV, 3.6Å, JEV, 4.3Å), the main interest for the broader virology community is the mechanism of the Fab inhibition. Indeed, high-resolution structure of virion-Fab complexes are more unusual (e.g. DEN2 and 3, 6.5Å and 9Å; Zika, 4-12Å). Unfortunately, the study does not clearly elucidate the mechanism of neutralization, showing some indication of inhibition of both receptor-binding and fusion.

The mechanistic aspects are less strongly supported than the first part of the manuscript. In particular, two points require additional data or analyses:

1. Two additional pairs of histidine residues are proposed to contribute to the pH-sensitive switch. While somewhat supported by the inter-histidine distance, one would expect more detailed analyses (pKa calculations; modelling) if not mutagenesis. Support from the previously published mutagenesis is tenuous.

A: To address the reviewer's comments, we performed pKa calculations for the amino acids at the E and M-protein interfaces using the Rosetta-pKa tool (Kilambi, 2012). (An identical approach was used previously for the dengue virus (Chaudhury, 2015).) Our results show that the histidines proposed to act as pH sensors/switches become protonated at a pH similar to that in the late endosomes (pH 5.8). In addition, we generated electrostatic surface potential maps of the 2x(E-M) hetero-tetramer at neutral and low pH to clearly show the changes in surface potential that occur during the pH change in endosomes (Fig. 2g). These results have now been included in the manuscript as Fig. 2g and discussed in the text (lines 200-214):

"Four out of the six histidines that interact with each other within the TBEV heterotetramer have pKa values equal to or higher than 5.8 (Fig. 2g). This provides additional evidence that the histidines could serve as pH sensors. These pKa calculations are sensitive to the precise locations of amino-acid side chains within the protein structure³⁵. It is therefore important that the role of histidines in controlling the pH-mediated conformational switch of the flavivirus E-proteins is supported by previous experimental evidence. Nelson et al. demonstrated that WNV E-proteins with single mutations in histidines do not differ from the wild-type virus in their capacity to induce membrane fusion³⁶. Similarly, Fritz et al. showed that single mutants of TBEV E-protein His248Asn and His287Ala could induce membrane fusion with an efficiency similar to that of the wild type³⁷. However, the double mutant His248Asn and His287Ala had a lower efficiency of formation of E-protein trimers, and its sub-viral particles were nearly fusion incompetent. In combination, the mutational and structural analyses provide evidence that the disruption of E-M heterodimers and detachment of the E-protein ectodomains from the virion membrane may depend on the protonation of histidines in the low pH of late endosomes."

The method used for the pKa calculations is reported in the Materials and methods section (lines 510-512)"

“Predictions of pKa were calculated using Rosetta-pKa³⁵ according to the method described for DENV E-protein by Chaudhury et. al⁶⁹. Electrostatic surfaces were visualized in the program PyMOL using the APBS and PDB2PQR plugins.”

2. Due to pleiomorphism, the acidic pH reconstruction has a low resolution and is possibly unreliable. Validation is again needed here to understand at what stage to the fusion might be blocked (e.g. compared to stages described in Chao et al. eLife 2014;3:e04389).

A: We agree with reviewer #1 that the pleiomorphism of TBEV virions in complex with Fab fragments at low pH leads to a reconstruction that is of limited resolution and should not be over-interpreted. Therefore, we limit our conclusions to the aspects of the reconstruction that are also supported by other analyses (e.g. the detachment of E-proteins from the virus membrane that is corroborated by the absence of the E-protein ring in 2D classes of the particles). FSC curve of the low pH TBEV-Fab complex reconstruction has now been included in Supplementary Fig 6a.

In order to more precisely determine the stage of fusion blocked by the antibody, we performed additional cell-fusion inhibition experiments (Fig. 5b). When the TBEV is mixed with IgG 19/1786, the virus becomes incapable of inducing cell fusion. However, in complex with Fab fragments of the antibody the cell fusion is only partly blocked. Because of the endpoint nature of the experiments (cryoEM reconstruction, and C6/36 cell experiment) we cannot identify the exact stage of the fusion blockage. We discuss our results with respect to the previous analyses performed by Chao et al. (lines 323-334):

“Particles with bound Fab fragments at low pH lack the density corresponding to the ectodomain layer, whereas the lipid bilayer enclosing the nucleocapsid core is intact (Fig. 5c,d). Notably, the leaflets of the lipid layer are spherical and have lost the deformations present in the native particles (Fig. 5c,d). This indicates a reorganization of the positions of transmembrane helices of the E and M proteins. Most likely the proteins lost their icosahedral ordering and became irregularly distributed in the virus membrane. Even though the ectodomains of E-proteins detached from the virus membrane, the fusion capability of the virus became impaired because of the Fab binding. Chao et al. had shown that availability of competent monomers within the contact zone between virus and target membrane makes trimerization a bottleneck in hemifusion¹¹. It is therefore possible that the Fab 19/1786 binding interferes with the conformational rearrangement of the E-protein dimers into fusogenic trimers.”

Overall, the manuscript presents useful structural details of the TBEV organisation with and without a neutralising Fab. However, in the absence of experimental data directly addressing the mechanism of pH-induced reorganisation and Fab inhibition, the advances may seem incremental outside of the flavivirus field compared to results established for DENV, WNV or Zika virus.

Further comments:

1. It is surprising that the following two articles are not discussed:

- Rouvinski A et al. Nature 2015;520(7545):109–13.

- Zhao H et al. Cell 2016;166(4):1016–27.

A: Thank you. We have now included the proposed articles in our discussion (lines 247-249):

“It is notable that antibodies E16 and ZV-54/ZV-67, which can neutralize WNV⁴⁰ and ZIKV⁴¹ respectively, also interact with the domain IIIs of E-proteins and bind to the virus particles with the same stoichiometry as that of 19/1786.”

And (lines 338-341):

“The crystal structure of a human antibody which is active against all DENV serotypes in complex with the E-protein ectodomain revealed an “E-dimer-dependent epitope” that includes the conserved main chain of the fusion loop and the two conserved glycosylation sites of the virus⁵².”

2. At this resolution, outliers in the Ramachandran plot need to be corrected or their validity discussed.

A: We re-refined the structure with tighter geometry restraints and thus reduced the number of Ramachandran outliers to 0.3%. Table 1 has been updated accordingly (page 30, lines 703-705).

3. The resolution of the constant domain of the Fab is surprisingly low. Have lower symmetry reconstructions been attempted? Also, there is no mention of movie-refinement or particle polishing. This is likely to further improve the quality of the resulting maps.

A: We tested several approaches to improving the resolution of the Fab constant regions: (1) Reconstruction with lower symmetry. (2) Tight masking the map to make a focused reconstruction. (3) Localized sub-particle reconstruction. None of the above attempts improved the quality of the map of the Fab constant region. We applied a movie alignment of frames before the reconstruction. Movie-refinement and refinement of particles from pre-aligned micrographs resulted in very similar reconstruction, as judged by FSC and visual inspections of the resulting maps. We have now included a more detailed description of the reconstruction process in the Materials and methods section (lines 456-471):

“For the reconstruction of the TBEV-Fab complex at low pH, single non-overlapping particles were boxed from the micrographs. Reference-free 2D classification was used to remove damaged particles. The TBEV structure low-pass filtered to 60 Å was used as an initial model. The particles were subjected to 3D classification, however this approach did not lead to a single class of uniform particles, but instead in each iteration the particles redistributed randomly among the three generated classes. Thus, all the particles that passed the 2D classification were used for the reconstruction process, which did not lead to a high-resolution map. We repeated the reconstruction with C1, C5, and icosahedral symmetries as well as with masks of different sizes in an attempt to remove the most pleiomorphic parts of the particles from the orientation determination process. The best results were achieved by masking out the region including the Fab fragments, and aligning the particles only according to the features of the underlying TBEV particle. The dataset was homogenized by 3D classification that used the orientations of the particles from the previous reconstruction. This approach partially eliminated some of the variability in the region containing the Fab fragments. The final reconstruction based on 5903 particle images was calculated using RELION.”

4. A 3-D representation of sequence conservation (e.g. ConSurf) around the histidine residues discussed in the text would be useful to complement Figure S3.

A: According to the reviewer’s suggestion, we have now included ConSurf visualization (Landau, 2005) of the residue conservation in the E- and M-protein of TBEV in the manuscript (Fig. 2f, Supplementary Fig. 5). The visualization demonstrates that the histidines involved in pH sensing are conserved among flaviviruses. The corresponding discussion has now been included in the manuscript (lines 197-200):

“Homologues of His287 and His419 of TBEV are present in several other flaviviruses, and the mechanism for inducing detachment of the E-protein ectodomain from the virus membrane might be shared within the virus family (Fig. 2f, Supplementary Fig. 5).”

References

- Chao, L. H., Klein, D. E., Schmidt, A. G., Peña, J. M. & Harrison, S. C. Sequential conformational rearrangements in flavivirus membrane fusion. *Elife* **3**, e04389 (2014).
- Chaudhury, S., Ripoll, D. R. & Wallqvist, A. Structure-based pKa prediction provides a thermodynamic basis for the role of histidines in pH-induced conformational transitions in dengue virus. *Biochem. Biophys. Reports* **4**, 375–385 (2015).
- Kilambi, K. P. & Gray, J. J. Rapid calculation of protein pKa values using rosetta. *Biophys. J.* **103**, 587–595 (2012).
- Landau, M. *et al.* ConSurf 2005: The projection of evolutionary conservation scores of residues on protein structures. *Nucleic Acids Res.* **33**, 299–302 (2005).

Reviewer #2 (Remarks to the Author):

In this study, Fuzik et al. report the cryo-EM structure of the TBEV virion alone and when bound by the monoclonal antibody 19/1786. While prior publications have reported the crystal structure of the TBEV E protein, and the cryo-EM structure of a TBEV subviral particle, this represents the first structure of the TBEV virion. As expected, the virion structure was found to be very similar to other solved Flavivirus virion structures (WNV, DENV, etc.). A comparison of the E proteins in the cryo-EM structure to the previously determined crystal structure elucidated a bend in the protein that reflects its orientation in the context of the spherical virion, as opposed to a monomer in solution. A detailed analysis of the binding characteristics of the monoclonal antibody 19/1786 supports binding of this antibody at the 3- and 5-fold axes of symmetry, but not the 2-fold. A particularly novel aspect of this study is the finding that the antibody footprint binds distinct domains at each of these sites. While the epitope is primarily focused on E domain III, when bound near the 3-fold symmetry axis the antibody also contacts residues in domain I, and when bound near the 5-fold symmetry axis the additional contacts are found in domain II.

Mutagenesis studies could be particularly interesting- can mutations in the antibody footprint of domain I or II inhibit binding of the antibody at one symmetry axis, but not the other? Is only one of these binding footprints responsible for inhibiting fusion?

A: As suggested by reviewer #2 we designed double- and quadruple-mutants in the antibody footprints within domains I and II. The mutations were selected to interfere with antibody binding. The neutralizing activity of the IgG 19/1786 was reduced from EC₅₀ 0.16µg/ml to 0.29-0.77µg/ml for the mutants. The limited effect of the mutations on the neutralizing activity of the antibody corroborates the structure-based observation that the roles of domains I and II in IgG 19/1786 binding are minor relative to the interactions with domain III. The results of the mutational and neutralization analyses and their discussion have now been included in the manuscript (Fig. 3c, text lines 274-299):

“To determine whether the interaction of antibody 19/1786 with domains I and II had any role in the virus neutralization, we prepared viruses with mutations in the antibody binding sites. In domain I two amino acids that form hydrogen bonds with the heavy chain of the antibody (Supplementary Fig. 7) were mutated to amino acids with opposite charges (double mutant Glu51Gln and Lys161Asn). To further disrupt the interaction interface, a quadruple mutant was prepared in which two additional amino acids with small side-chains were replaced with tyrosine (Glu51Gln, Lys161Asn, Ser158Tyr, and Thr279Tyr). In order to disrupt the binding site of the antibody in domain II two and four amino acids were replaced with residues with opposite charges (double mutant Asp67Asn, Lys69Glu and quadruple mutant Asp67Asn, Lys69Glu, Glu84Gln, and Gln87Glu). The mutations in its domain I made the virus more resistant to antibody 19/1786 neutralization, with EC₅₀ values increasing from 0.16 µg/ml for the wild-type virus to 0.77 µg/ml and 0.49 µg/ml for the double and quadruple mutants, respectively (p-values <0.0001; Fig. 3c). The quadruple mutant exhibited lower resistance to the neutralization than the double mutant (p=0.0068). Presumably the additional mutations in its domain I served as suppressor mutations instead of having the expected synergic effect. A less pronounced reduction in the neutralization activity was observed for the mutants of domain II, with EC₅₀ values increasing to 0.30 µg/ml and 0.29 µg/ml for the double and quadruple mutants, respectively (p-values ≤ 0.0086, Fig. 3c). The difference in the EC₅₀ values of the two mutants in domain II is not statistically significant (p=0.835). The results of these mutational experiments indicate that the interaction of antibody 19/1786 with domains I and II contributes to virus neutralization. The binding of the antibody to domain II may prevent the induction of membrane fusion, whereas the interaction with domain I may interfere with the hinge movement that is required for the formation of pre-fusion trimers⁸.

Nevertheless, the interactions of antibody 19/1786 with domains I and II appear to be only auxiliary, and the function of the antibody probably depends mostly on its interaction with domain III.”

Descriptions of the preparation of the footprint mutants and neutralization experiments have now been included in the Materials and methods section (lines 398-414):

“Recombinant TBEV (Oshima 5-10 strain) was prepared from infectious cDNA clones, Oshima-IC as reported previously⁵⁴. To introduce mutations, cDNA fragments with the mutations were synthesized by standard fusion-PCR and subcloned into Oshima-IC in a stepwise manner. Infectious RNA was transcribed from Oshima-IC using mMACHINE SP6 (Thermo Fisher) and transfected into BHK-21 cells using TransIT-mRNA (Mirus Bio LLC), as described previously⁵⁴. Recombinant viruses were recovered from cell culture supernatants.

To measure the ability of antibody 19/1786 to neutralize the TBEV, the mutant viruses were incubated with serially diluted antibody and inoculated into BHK-21 cells. The cells were grown in minimal essential medium containing 1.5 % carboxymethyl cellulose and 2 % FBS for 4 days. After 4 days of incubation, the cells were fixed with 10 % formalin and stained with 0.1 % crystal violet. Plaques were counted and expressed as plaque-forming units (PFU)/mL, and the reduction in the number of plaques by the antibody was evaluated. The neutralization experiments were done in triplicates. Neutralization curves were constructed from the measured data and EC₅₀ values with corresponding standard errors were calculated from the fitted Hill dose-response curve. For statistical comparison of the means of groups, an unpaired t-test was used and the corresponding P-values are reported.”

Overall, the manuscript is very well written, and the figures are clear. One piece of data that should be included is more information on the neutralizing activity of antibody 19/1786, including a neutralization curve/ experiment, as well as the EC₅₀ concentration. The reference to the antibody is a paper from 1994 that is not easily accessible to readers, even those with broad journal access.

A: As suggested by reviewer #2, we re-measured the neutralizing activity of the mAb IgG 19/1786 as well as the Fab 19/1786 (Fig. 3a,b). For the whole antibody, we measured a very similar EC₅₀ of 0.24µg/ml to that determined by Niedrig et al. in 1994 (EC₅₀ 0.2µg/ml). For the Fab fragment, the EC₅₀ concentration was 35µg/ml, almost 150 times higher than that of the whole antibody. This result is not unusual for Fab fragments (Plevka, 2014). These results have now been included in the manuscript (Fig. 3a,b, text lines 220-224):

“Mouse monoclonal antibody IgG1 19/1786 has therapeutic potential because it neutralizes multiple strains of TBEV and has minimal cross-reactivity with other flaviviruses³⁸. We determined the EC₅₀ values for the whole antibody and Fab fragment to be 0.24±0.03 and 35.0±2.5 µg/ml, respectively (Fig. 3a,b). It is common that the inhibiting concentration of Fab is more than 100 times higher than that of the full antibody³⁹.”

The neutralization assay is described in the Materials and methods section (lines 375-383):

“Virus neutralization by the 19/1786 mAb and corresponding Fab fragments was measured according to a previously published protocol³⁸. Briefly, serial dilutions of the antibody and Fab fragment were prepared, mixed with TBEV (1,000 pfu/ml), and incubated at 37°C for 2 h. After the incubation, the mixtures were applied to monolayers of porcine kidney stable cells in 96-well plates, and incubated for 4 days at 37°C. The cytolysis was examined using light microscopy and the neutralization rate was determined. The assay was done in triplicates for the IgG and in duplicates for the Fab. Neutralization curves were constructed from the measured data and EC₅₀ values with corresponding standard errors were calculated from the fitted Hill dose-response curve.”

Additional comments include:

Lines 31-34. The authors state in the abstract “We show that the repulsive interactions of histidine side chains, which are protonated at low pH, disrupt heterodimers of TBEV envelope and membrane proteins and induce detachment of the E protein ectodomains from the virus membrane.” However, this is not experimentally shown, only hypothesized based on the proximity of specific histidine residues in the virus structure. Furthermore, a prior study in WNV did not support the histidine switch hypothesis (PMID: 19776132). The wording should be adjusted to not overstate the findings.

A: We performed additional analyses to show that the His side chains in TBEV are probably charged at low pH (please see our answer to Reviewer #1 comment 1). We adjusted our statement about the proposed His switch according to the reviewer’s suggestion (lines 34-38):

“The virion structure indicates that the repulsive interactions of histidine side chains, which become protonated at low pH, may contribute to the disruption of heterotetramers of the TBEV envelope and membrane proteins and induce detachment of the envelope protein ectodomains from the virus membrane.”

Lines 120-122. “The E-proteins of TBEV, WNV, ZIKV, and Japanese encephalitis virus contain a single homologous glycosylation site, whereas, that of DENV has an additional glycosylation site at Asn67.” While true that the majority of these virus strains are glycosylated, there are strains of WNV that are not glycosylated, and perhaps other viruses as well. I suggest changing this to say “most”, or “the majority”.

A: Thank you, we accept the correction suggested by reviewer #2 (lines 126-128):

“The E-proteins of the majority of TBEV, WNV, ZIKV, and Japanese encephalitis virus strains contain a single homologous glycosylation site, whereas that of DENV has an additional glycosylation site at Asn67⁸.”

Supplementary Figure 3. Please indicate the strain of WNV (I assume NY99). The symbols denoting conservation versus absolute conservation are incorrect based on the sequence alignment. “*” should be absolute conservation, and “:” conservation. In the sequence alignment, WNV is written incorrectly as WNW.

A: We have now included the WNV strain identifier “956” in the figure (now Supplementary Fig. 5). We corrected the figure legend according to reviewer’s suggestion – Thank you. (Lines 842-845):

“Conservation of a residue is denoted as follows: “*” - absolute conservation; “:” - conservation of amino acids with strongly similar properties; “” – conservation of amino acids with weakly similar properties.”

References

- Fritz, R., Stiasny, K. & Heinz, F. X. Identification of specific histidines as pH sensors in flavivirus membrane fusion. *J. Cell Biol.* **183**, 353–361 (2008).
 Plevka, P. et al. Neutralizing antibodies can initiate genome release from human enterovirus 71. *Proc. Natl. Acad. Sci.* **111**, 2134–2139 (2014).
 Zhang, X. et al. Cryo-EM structure of the mature dengue virus at 3.5-Å resolution. *Nat. Struct. Mol. Biol.* **20**, 105–110 (2013).

Reviewer #3 (Remarks to the Author):

The cryo-EM structures are presented for the native TBEV virion and TBEV in complex with Fab fragments of neutralizing antibody 19/1786. The finding is novel because instead of locking the virus E-proteins into the native-like state, the authors believe the structure shows that repulsive interactions of histidine side chains disrupt heterodimers of TBEV envelope/membrane proteins to cause detachment of the E protein ectodomains from the virus membrane.

Minor issues: stylistic problems include:

1) Details in the abstract (repulsive interactions of histidine side chains) that might be better swapped with the lack of detail in the final paragraph of the intro (binding of Fab fragments does not prevent low-pH-induced movements of E proteins, but blocks membrane fusion - however, blocking membrane fusion is stated as conclusive, whereas it is suggestive).

A: To address the reviewer's comments, we modified both the abstract and the end of the introduction in the following ways:

We softened our statements in the abstract (lines 34-41):

"The virion structure indicates that the repulsive interactions of histidine side chains, which become protonated at low pH, may contribute to the disruption of heterotetramers of the TBEV envelope and membrane proteins and induce detachment of the envelope protein ectodomains from the virus membrane. The Fab fragments bind to 120 out of the 180 envelope glycoproteins of the TBEV virion. Unlike in the previously studied flavivirus-neutralizing antibodies, the Fab fragments do not lock the E-proteins in the native-like arrangement, but interfere with the process of virus-induced membrane fusion."

We re-wrote the final paragraph of the introduction to describe the proposed His switch and removed the conclusive statement about the inhibition of membrane fusion (lines 84-90):

"Here, we report the structures of the native TBEV virion and its complex with the Fab fragments of the neutralizing antibody 19/1786. Our results indicate that the low-pH induced protonation of histidines may contribute to disruption of the E-M heterotetramers and induce detachment of the E-protein ectodomains from the virus membrane. Furthermore, the binding of 19/1786 antibodies to the TBEV surface does not prevent the low-pH-induced movements of E-proteins, however, it does interfere with the virus-induced membrane fusion."

2) Excessive use of short sentences leading to a choppy presentation, especially in introduction.

A: We have re-written parts of the introduction to make the text more continuous. Please see lines 42-83.

Line 100 - is the correct figure called here: "icosahedral asymmetric unit form unique interactions with the surrounding glycoproteins (Fig. 1d)"

A: Fig. 1d (Fig. 1c after revision) shows that TBEV asymmetric unit consists of three E-proteins that form distinct interactions with each other. We edited the legend of Fig. 1c to emphasize this interpretation of the figure (lines 712-716):

"(c) Molecular surface of TBEV virion low-pass filtered to 7 Å. The three E-protein subunits within each icosahedral asymmetric unit are shown in red, green, and blue. The three E-proteins in the icosahedral asymmetric unit form unique interactions with each other (for more detail see Supplementary Fig. 2). The black triangle shows the borders of a selected icosahedral asymmetric unit."

Furthermore, we have now included Supplementary Fig. 2, which shows details of all non-quasi-equivalent E-protein contacts.

The authors are referring back to the central section of the map (Fig 1-d) to describe inner and outer leaflets, separation, shape, insertion, core shape etc without indicating any of these features in the figure. A color coded panel added to figure 2 might provide clarity and prevent the reader from shuffling back and forth between fig 1, 2, and 3 to follow the description.

A: We have now re-arranged and re-numbered figures to reduce cross-referencing in the text. In addition we have now synchronized the color-coding in Figs. 1d and 5c,d, Supplementary Fig. 6c to clarify the organization of the flavivirus particle.

Figure 1

Scale bars in d and e are same (10 nm), yet the surface rendered 'd' map is clearly larger than b or e.

A: Thank you. We have modified Fig. 1 according to the reviewer's suggestion. Now all panels showing reconstructions of the TBEV particle have the same scale.

Add arrow to b and e to indicate location of E and M-proteins.

A: Instead of adding arrows, we color-coded a portion of the TBEV particle in panel 1d (new numbering) to distinguish the E and M proteins. The color-coding is described in the figure legend (lines 716-719):

“(d) Central slice of TBEV electron density map perpendicular to the virus fivefold axis. The virus membrane is deformed by the transmembrane helices of E and M-proteins. The lower right quadrant of the slice is color-coded as follows: nucleocapsid – blue; inner and outer membrane leaflets – orange; M-proteins – red; E-proteins – green.”

The color-coding is synchronized among Figs. 1d and 5c,d, Supplementary Fig. 6c.

M-proteins are poorly visible in Fig. 1b (new numbering), which displays an isosurface map of the TBEV particle. We prefer not to add arrows to this section of the figure.

The legend does not indicate if the map displayed as 'b' is a sharpened map

A: We corrected the figure legend to state that the map was sharpened (lines 709-711):

“(b) B-factor sharpened electron-density map of TBEV virion, rainbow-colored according to distance from particle center. The front lower-right eighth of the particle was removed to show the transmembrane helices of E and M-proteins.”

Figure 1 and 2

Keeping the capsid diameter the same between figures and panels would add clarity. Please address and define in the legend the difference between the rendering the maps used in Fig 1-a, 1-d, 2-b, 2-c, 3-a, 3-d. Some of these appear high resolution and some do not.

A: We have now combined Figs. 1 and 2 (new Fig. 1). Both panels of Fig. 1 showing reconstructions of the TBEV virion (b and c) now have the same scale. The capsid diameters are kept the same also for Supplementary Fig. 1. We have now extended the figure 1 legends to clearly state what type of electron density is being displayed (lines 709-716):

“(b) B-factor sharpened electron-density map of TBEV virion, rainbow-colored according to distance from particle center. The front lower-right eighth of the particle was removed to show the transmembrane helices of E and M-proteins. (c) Molecular surface of TBEV virion low-pass filtered to 7 Å. The three E-protein subunits within each icosahedral asymmetric unit are shown in red, green, and blue. The three E-proteins in the icosahedral asymmetric unit form unique interactions with each other (for more detail see Supplementary Fig. 2). The black triangle shows the borders of a selected icosahedral asymmetric unit.”

The local resolution map is difficult to interpret as rendered and the legend does not describe the local resolution map adequately. Is this a cut-away half map? Or is it a slabbed cross-section with noise shown in grey in the center?

A: The local resolution map (now Supplementary Fig. 1c) is a cut-away half map color-coded according to the estimated local resolution. The best resolved parts are shown in deep blue while the worse-resolved parts gradually change colors through green, yellow and red. Features with a resolution worse than 7 Å are shown in grey and mostly appear as noise in the central part of the map. We have now extended the descriptions of the local resolution maps in the legends of Supplementary Fig. 1c (and Supplementary Fig. 6b that shows local resolution of TBEV-Fab complex) (lines 800-805).

“(c) Local resolution of cryo-EM map of TBEV virion. The display shows a cut-away half map colored according to the local resolution. The best resolved rigid parts include the ectodomains of the E-proteins. In contrast the virus membrane was reconstructed with less detail. Parts of the map with resolution worse than 7 Å are shown in grey. The non-sharpened electron density map was used for the display.”

Figure 4

The coloring in panel c is helpful to orient the viewer, but the figure would be improved if the maps in b and c were the same diameter and displayed adjacent to each other evenly. Again the change (between b and c) in contour or sharpening needs to be included in the legend.

A: We unified the scale of the particles displayed in panels b and c of Fig. 4. We have now included information about the sharpening of half of the map in panel b in the figure legend (lines 758-761):

“(b) Electron-density map of Fab-covered TBEV virion rainbow-colored according to distance from center of particle. The right half of the image represents a B-factor sharpened map. Electron densities corresponding to the Fab fragments are located close to the threefold and fivefold symmetry axes of the virion.”

Furthermore, we have now extended a description of panel c to state that it represents a low-pass filtered molecular surface map of the atomic model (lines 761-764):

“(c) Molecular surface of TBEV virion covered with Fab 19/1786 fragments low-pass filtered to 7 Å resolution. E-proteins are shown in red, green, and blue. Fab fragments are shown in magenta (heavy chain) and pink (light chain).”

Fig 6 is described by one line (220) and may be better suited to supplemental.

A: According to the reviewer’s suggestion, we have now moved Fig. 6 to become Supplementary Fig. 10.

Line 243 is overstated, finding in the fab footprint a residue implicated by another work to be important to receptor binding does suggest the Fab binding might block receptor, but it is not as certain as stated.

A: We have modified the sentence to be more conditional (lines 268-273):

“It was shown previously that the replacement of Thr310 in the TBEV E-protein with another amino acid resulted in a decreased infectivity of the mutant virus⁴². It was speculated that this residue is important for receptor recognition⁴². Thr310 is part of the 19/1786 binding site (Supplementary Fig. 7,8). It is therefore possible that the interaction of antibody 19/1786 with domain III may interfere with the binding of TBEV to its putative receptor.”

Complex at low pH -- there were ~5000 particles in the 20Å map, but no other statistics were provided. The raw image shows overlapping capsids. There might be other contributing factors

limiting resolution and the poor resolution of the map alone is not enough to allow some of the conclusions.

A: We extended the Materials and methods section to describe the process of the reconstruction of the TBEV Fab complex at low pH in detail. Our analyses indicate that the low resolution of the reconstruction is due to the pleiomorphic nature of the TBEV-Fab particles under low pH. FSC curve of the low pH TBEV-Fab complex reconstruction has now been included in Supplementary Fig 6a. Lines 456-471:

“For the reconstruction of the TBEV-Fab complex at low pH, single non-overlapping particles were boxed from the micrographs. Reference-free 2D classification was used to remove damaged particles. The TBEV structure low-pass filtered to 60 Å was used as an initial model. The particles were subjected to 3D classification, however this approach did not lead to a single class of uniform particles, but instead in each iteration the particles redistributed randomly among the three generated classes. Thus, all the particles that passed the 2D classification were used for the reconstruction process, which did not lead to a high-resolution map. We repeated the reconstruction with C1, C5, and icosahedral symmetries as well as with masks of different sizes in an attempt to remove the most pleiomorphic parts of the particles from the orientation determination process. The best results were achieved by masking out the region including the Fab fragments, and aligning the particles only according to the features of the underlying TBEV particle. The dataset was homogenized by 3D classification that used the orientations of the particles from the previous reconstruction. This approach partially eliminated some of the variability in the region containing the Fab fragments. The final reconstruction based on 5903 particle images was calculated using RELION.”

Line 270, overstatement that the ectodomains of E proteins detached from the virus membrane, but due to the Fab binding could not induce membrane fusion.

A: We have now modified the sentence to avoid the possibility of over-interpretation of our results (lines 328-334):

“Most likely the proteins lost their icosahedral ordering and became irregularly distributed in the virus membrane. Even though the ectodomains of E-proteins detached from the virus membrane, the fusion capability of the virus became impaired because of the Fab binding. Chao et al. had shown that availability of competent monomers within the contact zone between virus and target membrane makes trimerization a bottleneck in hemifusion¹¹. It is therefore possible that the Fab 19/1786 binding interferes with the conformational rearrangement of the E-protein dimers into fusogenic trimers.”

There is not enough evidence presented here to conclude absolutely that the Fab fragments inhibited the membrane fusion process. However, the comparisons presented in figure 7 are suggestive. This presentation would be helped by a color-coded map (see above) to help the reader understand the three distinct layers representing which are the inner and outer leaflets of the membrane and the ectodomain of the E-proteins.

A: According to the reviewer’s suggestion, we have now color-coded the 2D class averages in Fig. 5c,d. Furthermore, we performed additional experiments to show that native TBEV efficiently mediates cell fusion at low pH, but the virus neutralized by IgG 19/1786 cannot induce the cell fusion (Fig. 5b). The results are described in the Results and discussion section (lines 323-331):

“Particles with bound Fab fragments at low pH lack the density corresponding to the ectodomain layer, whereas the lipid bilayer enclosing the nucleocapsid core is intact (Fig. 5c,d). Notably, the leaflets of the lipid layer are spherical and have lost the deformations present in the native particles (Fig. 5c,d). This indicates a reorganization of the positions of transmembrane helices of the E and M proteins. Most likely the proteins lost their icosahedral ordering and became irregularly distributed in the virus membrane. Even though the ectodomains of E-proteins

detached from the virus membrane, the fusion capability of the virus became impaired because of the Fab binding.”

And (lines 309-315):

“The binding of Fab fragments of antibody 19/1786 prevented the fusion of TBEV virions at pH 5.8 (Fig. 5a). However, this might be caused by the inaccessibility of the virus membrane at the surface of the Fab-decorated TBEV virions. To determine whether IgG 19/1786 and Fab 19/1786 can prevent membrane fusion *in vivo*, we performed a “fusion-from-without” assay using C6/36 cells⁴⁸. Whereas the native TBEV induces cell fusion at low pH, the virus in complex with IgG 19/1786 lost this ability, and the virus in complex with the Fab 19/1786 induced cell fusion with lower efficiency than the native virus (Fig. 5b).”

Description of the C6/36 cell fusion assay has been included in the Materials and methods section (lines 386-395):

“A fusion-from-without assay was performed as described previously⁵³. Mosquito C6/36 cells were grown in 96-well tissue cell culture plates for 2 days. The cells were precooled for 45 min at 4 °C, then washed with serum-free medium. Cells were incubated for 1 hour at 4 °C with 30 µl of purified virus at a concentration of 500 µg/ml or a mixture of virus pre-incubated (30 min) with IgG 19/1786 (100 µg/ml) or Fab 19/1786 (3,000 µg/ml). After removal of the virus suspension, pre-warmed fusion medium (MEM buffered with 20 mM MES, pH 5.5) was added to the cells and the plates were incubated for 2 min at 40°C. Fusion medium was replaced with a growth medium, the cells were further incubated at 40°C for 2 h, and then the cells were fixed with a 1:1 mixture of methanol and acetone and stained with Giemsa's solution.”

Insufficient details are provided for data collection in Methods.

A: We have now included additional information about data collection in the Materials and methods section (lines 423-437):

“To prepare the virus-Fab 19/1786 complex, TBEV particles were incubated with the Fab 19/1786 for 2 h at 4°C, using equimolar amounts of the Fab fragments and E-proteins. To study the mechanism of virus neutralization by Fab 19/1786, the pH of the sample was adjusted to 5.8 with 100 mM MES pH 5.5 and the sample was incubated for 15 min at 4°C. Samples for cryo-EM were vitrified using an FEI Vitrobot Mark IV on Quantifoil R2/1 grids with the following settings: 3.8 µl sample; wait time 10 s; blot time 2 s; blot force -2.

The grids with vitrified virions were loaded into an FEI Titan Krios microscope operating at 300kV, equipped with an FEI Falcon II direct electron detector. The microscope illumination and projection system was aligned before data acquisition, and the astigmatism and coma-free alignments were corrected every 12 h during the acquisition process. The micrographs were acquired using the automated acquisition software EPU (FEI) at defoci varying between 1-3 µm at 75,000x magnification, resulting in a pixel size of 1.063 Å. Six acquisition areas were defined per foil-hole and autofocus was performed before the acquisition of each foil-hole. Images were recorded as seven-frame movies, with a total exposure time of 0.5 seconds and dose of 22 e⁻/Å².”

Table 1 Ramachandran outliers 1.13% in virus map, is high

A: We re-refined the structure and reduced the number of Ramachandran outliers to 0.3%. Table 1 has been modified to show the new results (page 30, lines 703-705).

Reviewers' Comments:

Reviewer #1 (Remarks to the Author):

The authors have appropriately replied to the points raised regarding the structural analysis. The mechanistic aspects are not as conclusive as one may wish due to the absence of mutagenesis of histidine residues but the authors have toned down their statements accordingly.

The sentence "Unlike in the previously studied flavivirus-neutralizing antibodies..." in the abstract should be modified to reflect the similar mechanism of action of DV2-E104 Fab.

Reviewer #2 (Remarks to the Author):

The revised version of the manuscript contains significant edits and additional experiments to address the reviewer's suggestions. Two minor comments:

1. Lines 79-81: JEV structure has recently been solved as well (PMID: 28446752)
2. Figure 5b- the top left panel appears to be mislabeled, as I believe this image is cells only, without TBEV as stated.

Reviewer #3 (Remarks to the Author):

Images are improved and clarity has been added especially in methods and figure legends. Referee comments have been addressed throughout.

Response to reviewer's comments

Reviewer's comments are highlighted in blue italics, our responses in bold black text.

Reviewer #1 (Remarks to the Author):

The authors have appropriately replied to the points raised regarding the structural analysis. The mechanistic aspects are not as conclusive as one may wish due to the absence of mutagenesis of histidine residues but the authors have toned down their statements accordingly.

The sentence "Unlike in the previously studied flavivirus-neutralizing antibodies..." in the abstract should be modified to reflect the similar mechanism of action of DV2-E104 Fab.

A: We have now modified the abstract according to reviewer's suggestion (lines 35-37):

"Unlike most of the previously studied flavivirus-neutralizing antibodies, the Fab fragments do not lock the E-proteins in the native-like arrangement, but interfere with the process of virus-induced membrane fusion."

Reviewer #2 (Remarks to the Author):

The revised version of the manuscript contains significant edits and additional experiments to address the reviewer's suggestions. Two minor comments:

1. Lines 79-81: JEV structure has recently been solved as well (PMID: 28446752)

A: We have now added JEV to the list of flaviviruses with known structures (and the corresponding reference) (lines 76-78):

"The structures of mature virions of the dengue (DENV), Zika (ZIKV), West Nile viruses (WNV), and Japanese encephalitis virus (JEV) and of the sub-viral particle of tick-borne encephalitis virus were solved previously by cryo-EM^{6,7,22-24}."

2. Figure 5b- the top left panel appears to be mislabeled, as I believe this image is cells only, without TBEV as stated.

A: Thank you, we corrected the labeling in Figure 5b.

Reviewer #3 (Remarks to the Author):

Images are improved and clarity has been added especially in methods and figure legends. Referee comments have been addressed throughout.

A: Thank you.